# Immunoinformatics mapping of potential epitopes in SARS-CoV-2 structural proteins

Yengkhom Damayanti Devi[1☯], Himanshu Ballav Goswami[1☯], Sushmita Konwar[1☯], Chandrima Doley[1☯], Anutee Dolley[1☯], Arpita Devi[1], Chen Chongtham[1,2], Dikshita Dowerah[3], Vashkar Biswa[4], Latonglila Jamir[5], Aditya Kumar[1], Siddhartha Shankar Satapathy[6,7], Suvendra Kumar Ray[1,7], Ramesh Chandra Deka[3,7], Robin Doley[1], Manabendra Mandal[1,7], Sandeep Das[4], Chongtham Shyamsunder Singh[8], Partha Pratim Borah[9], Pabitra Nath[10], Nima D. Namsa[1,7] *

1 Department of Molecular Biology and Biotechnology, Tezpur University, Napaam, Assam, India, 2 National Institute of Immunology, Aruna Asaf Ali Marg, Jawaharlal Nehru University, New Delhi, India, 3 Department of Chemical Sciences, Tezpur University, Napaam, Assam, India, 4 Department of Biotechnology, Bodoland University, Kokrajhar, Assam, India, 5 Department of Environmental Science, Nagaland University (Central), Lumami, India, 6 Department of Computer Science and Engineering, Tezpur University, Napaam, Assam, India, 7 Centre for Multi-disciplinary Research, Tezpur University, Napaam, Assam, India, 8 Department of Paediatrics, Regional Institute of Medical Sciences, Imphal, India, 9 Department of Paediatrics and Neonatology, Pratiksha Hospital, Borbari, Guwahati, Assam, India, 10 Department of Physics, Tezpur University, Napaam, Assam, India

☯ These authors contributed equally to this work.
* namsa@tezu.ernet.in

**Data Availability Statement:** All the data associated with the manuscript is provided as S1 File.

## Abstract

All approved coronavirus disease 2019 (COVID-19) vaccines in current use are safe, effective, and reduce the risk of severe illness. Although data on the immunological presentation of patients with COVID-19 is limited, increasing experimental evidence supports the significant contribution of B and T cells towards the resolution of severe acute respiratory syndrome coronavirus 2 (SARS-CoV-2) infection. Despite the availability of several COVID-19 vaccines with high efficacy, more effective vaccines are still needed to protect against the new variants of SARS-CoV-2. Employing a comprehensive immunoinformatic prediction algorithm and leveraging the genetic closeness with SARS-CoV, we have predicted potential immune epitopes in the structural proteins of SARS-CoV-2. The S and N proteins of SARS-CoV-2 and SARS-CoVs are main targets of antibody detection and have motivated us to design four multi-epitope vaccines which were based on our predicted B- and T-cell epitopes of SARS-CoV-2 structural proteins. The cardinal epitopes selected for the vaccine constructs are predicted to possess antigenic, non-allergenic, and cytokine-inducing properties. Additionally, some of the predicted epitopes have been experimentally validated in published papers. Furthermore, we used the C-ImmSim server to predict effective immune responses induced by the epitope-based vaccines. Taken together, the immune epitopes predicted in this study provide a platform for future experimental validations which may facilitate the development of effective vaccine candidates and epitope-based serological diagnostic assays.

**Funding:** The author(s) received no specific funding for this work.

**Competing interests:** The authors have declared that no competing interests exist.

## Introduction

Coronavirus disease 2019 (COVID-19) is a highly transmissible acute respiratory disease caused by a novel strain of coronavirus called the severe acute respiratory syndrome coronavirus 2 (SARS-CoV-2). Phylogenetic analysis of whole-genome sequences of SARS-CoV-2 isolated from infected patients revealed an overall sequence identity of 96.2%, 79.6%, and 50% with the genome of RaTG13, SARS-CoV BJ01, and MERS-CoV, respectively [1, 2]. The genome of SARS-CoV-2 encodes both structural and nonstructural proteins (NSPs). The first ORFs (ORF1a/b) encode 16 NSPs (NSP1-16) except in gamma coronavirus which lacks NSP1. There is a -1 frameshift between ORF1a and ORF1b leading to the production of two polypeptides (pp1a and pp1ab). These polyproteins are post-translationally processed by virus-encoded chymotrypsin-like protease (3CLpro) or main protease (Mpro) and by one or two papain-like proteases into 16 NSPs. The structural proteins include spike glycoprotein (S), envelope protein (E), membrane protein (M), and nucleocapsid protein (N). The S protein has two functional subunits that mediate cell attachment (S1 subunit) and virus-host membrane fusion (S2 subunit). The S proteins of SARS-CoV-2 and SARS-CoV are phylogenetically closely related with an amino acid sequence identity of approximately 77% [1] while utilizing the same cellular receptor angiotensin-converting enzyme 2 (ACE2) for entry into cells [3].

Antibodies that bind to the S protein have been reported to neutralize SARS-CoV-2. The ensuing rapid development of neutralizing antibodies against the S protein is correlated with the immune response to the virus, and individuals who show seroconversion may develop a lasting immune response against SARS-CoV-2 [4–6]. In a recent study, a rapid diagnostic test for the serodiagnosis of COVID-19 was developed using the S and N proteins of SARS-CoV-2 [7–9]. Many studies have reported the generation of IgM- and IgG-specific neutralizing antibodies against the S protein of SARS-CoV-2 [10–15]. Limited information is available on specific SARS-CoV-2 proteins that are recognized by immune cells; however, a few studies have found a higher proportion of T-cell responses specific to structural proteins in convalescent serum from patients with COVID-19 [16–18]. As of June 3, 2021, the World Health Organization (WHO) has approved the emergency use of COVID-19 vaccines, namely AstraZeneca/Oxford, Moderna, Johnson and Johnson, Pfizer/BioNTech, Sinopharm, and Sinovac Biotech after the evaluation of safety and efficacy data from clinical trials. The findings from clinical trials of a Chimpanzee adenoviral vector vaccine (ChAdOx1 nCoV-19, AZD1222) expressing the S protein of SARS-CoV-2 protein demonstrated an acceptable safety profile and good efficacy in symptomatic patients [19]. B.1.1.7, a new variant of SARS-CoV-2, has emerged as the dominant variant of COVID-19 in the UK. The clinical efficacy of AZD1222 against symptomatic infection was 70·4% for the B.1.1.7 variant and 81·5% for non-B.1.1.7 lineages; however, the neutralization activity of AZD1222 was lower against the B.1.1.7 variant than that against the non-B.1.1.7 variants [20]. The Moderna COVID-19 (mRNA-1273) vaccine is a lipid nanoparticle-encapsulated nucleoside-modified mRNA vaccine, which encodes the stabilized prefusion S protein of SARS-CoV-2. It showed an overall efficacy of 94.1% (95% confidence interval = 89.3–96.8%) in preventing symptomatic patients diagnosed with COVID-19, including those with severe symptoms [21]. The vaccine was also highly effective in clinical trials, exhibiting a high efficacy in individuals with different demographical characteristics, including age, sex, race, and ethnicity, as well as in individuals with pre-existing medical conditions. In a phase 1 trial of the mRNA-1273 vaccine, serum neutralizing antibody responses were detected in all participants (40 adults aged between 56 to 70 years), which were similar to those previously reported among vaccine recipients by multiple methods [22]. The vaccine elicited an increased production of CD4 response lymphocytes, including type 1 helper T cells [22]. The COVID-19 vaccine developed by Johnson and Johnson's/Janssen (Ad.26.COV2.S) is a

recombinant, replication-incompetent adenovirus serotype 26 (Ad26) vector vaccine, which encodes the stabilized prefusion S protein of SARS-CoV-2. The Johnson and Johnson vaccine showed 66.3% efficacy in symptomatic patients with laboratory-confirmed COVID-19 with no prior history of COVID-19 [23]. The efficacy of Ad.26.COV2.S, which varied geographically, was the highest in the United States (74.4%; 95% CI = 65.0–81.6%), followed by Latin America (64.7%; 95% CI = 54.1–73.0%) and South Africa (52.0%; 95% CI = 30.3–67.4%) [23]. These findings indicated that CD4+ T-cell responses were observed in 76 to 83% of the participants in cohort 1. In addition, 60 to 67% of those in cohort 3 showed a clear skewing toward type 1 helper T-cell responses on day 14 post-vaccination [23]. The Pfizer-BioNTech COVID-19 (BNT162b2) vaccine is a lipid nanoparticle-formulated nucleoside-modified mRNA vaccine, which encodes the prefusion S protein of SARS-CoV-2. Randomized clinical trials of a two-dose regimen of BNT162b2 revealed an overall efficacy of 95% in patients with COVID-19 who were 16 years of age or older [24]. The Sinopharm COVID-19 or BIBP is an inactivated vaccine developed by Beijing Bio-Institute of Biological Products Co Ltd. This vaccine showed an overall efficacy of 79% in preventing symptomatic disease and hospitalization [25]. Sino-vac-CoronaVac is an inactivated vaccine developed by Sinovac Biotech showed efficacies of 51% against symptomatic SARS-CoV-2 infection, 100% against severe COVID-19, and 100% against hospitalization in the studied population [26]. Although the above-mentioned vaccines are effective against both symptomatic and asymptomatic COVID-19, the recent global rise in the emergence of SARS-CoV-2 variants might compromise the effectiveness of these vaccines. In recent years, *in silico* vaccine designing, which involves the prediction of immunogenic antigenic sequences in viral proteins, has gained increasing attention due to advantages such as the rapid screening and identification of multiple antigen candidates that can generate an immune response *in vivo*. This approach harnesses high-throughput proteomics data available in public databases to screen for the most effective candidate epitopes based on criteria such as antigenicity and immunogenicity profiles. Therefore, the identification of B-cell and T-cell epitopes of structural proteins is essential for developing effective diagnostic tests and epitope-based vaccine candidates against SARS-CoV-2. Although computational approaches have successfully predicted potential B- and T-cell epitopes, a detailed analysis of epitopes of four structural proteins of SARS-CoV-2 might have important implications in the design and analysis of COVID-19 vaccines under various stages of clinical development [27–38]. Currently, well-developed bioinformatic approaches for epitope analysis have been used to successfully identify epitopes that generate both weak and strong immune responses, which might have been experimentally ignored [39]. In our study, we used multiple online bioinformatics resources and stringent selection criteria to identify potent B- and T-cell epitopes of four structural proteins of SARS-CoV-2. Our *in-silico* prediction method has led to the identification of potent, common, and species-specific B- and T-cell epitopes, which are likely to be recognized in humans. Moreover, we determined the conservancy of the predicted epitopes across different species of coronaviruses (CoVs).

## Materials and methods

### Sequence retrieval of structural proteins of SARS-CoV-2 and SARS-CoV

Coding sequences for structural proteins including S, N, M, and E proteins of SARS-CoV-2 (isolate WIV02, accession number MN996527.1) [1] and SARS-CoV (isolate BJ01, accession number AY278488.2) were retrieved from NCBI GenBank [40]. Sequence alignment of the receptor-binding domains (RBDs) of the S protein of SARS-CoV-2 (QHR63250.2) and SARS-CoV (QHR63300.2) was performed using Clustal Omega.

## Prediction of B-cell epitopes

Linear B-cell epitopes were predicted using BepiPred-2.0, Bcepred, and ABCpred online servers. The BepiPred-2.0 server predicts linear B-cell epitopes based on a web-based random forest algorithm, which is trained on epitopes annotated from antibody-antigen protein structures [41]. The Bcepred server predicts B-cell epitopes based on physicochemical properties such as hydrophilicity, flexibility, accessibility, polarity, exposed surface, number of turns, and antigenic propensity [42]. The ABCpred server uses an artificial neural network algorithm to predict linear B-cell epitopes in an antigen sequence with 65.93% accuracy. In this study, we applied a window length of 18 amino acids (aa) with a threshold setting of 0.7 and ranked the predicted B-cell epitopes according to their score; a higher score indicates a higher probability of being an epitope [43].

Moreover, EPSVR, DiscoTope, CBTOPE, and ElliPro servers were used to predict conformational B-cell epitopes. The EPSVR server uses a support vector regression (SVR) method to predict antigenic B-cell epitopes [44]. The DiscoTope server predicts discontinuous B-cell epitopes from 3D structures of proteins. This method calculates the surface accessibility and a novel epitope propensity amino acid score to predict potential epitopes [45]. The CBTOPE server predicts conformational B-cell epitopes from the primary sequence of the protein with an accuracy of more than 85% [39]. The ElliPro server predicts linear and discontinuous B-cell epitopes based on the protein structure and the homology-based model of the amino acid sequence [46]. The helical behavior of predicted monomeric peptides was computed using the Agadir server (http://agadir.crg.es) [47].

## Prediction of T-cell epitopes

TepiTool is an interactive and easy-to-use tool used to predict potential major histocompatibility complex (MHC) class I- and class II-binding peptides based on a panel of 27 most frequent alleles [48]. The Proteasomal cleavage/TAP transport/MHC class I combined predictor (http://tools.iedb.org/processing) in the Immune epitope database and analysis resource (IEDB) integrates data on proteasomal processing, transporter associated with antigen processing (TAP) transport, and MHC binding to produce an overall score for the intrinsic potential of each peptide being a T-cell epitope [49]. The MHC-NP predictor predicts T-cell peptides naturally processed by the MHC [50]. The NetMHCIIpan 4.0 server predicts peptide binding to known sequences of any MHC II molecule using artificial neural networks [51]. Three different tools, the IEDB combined server, TepiTool, and MHC-NP were used to predict 9-mer epitopes of MHC I. First, the 9-mer epitope list was generated using the 27 HLA-Class I allele reference list. Next, the top 2% were chosen from the high-scoring epitopes with an IC50 cut-off value of less than or equal to 500 nM. In TepiTool, MHC-I epitopes were searched from the query sequence using 'the panel of 27 most frequent A & B alleles' available in the server. In the 'Prediction method', the default IEDB method was selected. When the field 'peptides' was to be included in prediction, the default setting was applied at 'low number of peptides' since the epitope length was only 9 mer. Finally, the peptides were predicted based on the IC50 cut-off value of less than or equal to 500 nM. In MHC-NP, all default parameters were selected, and 9-mer epitopes that bind to 17 class I alleles were selected. The top 2% highest binders (epitopes with highest scores) were selected from the raw files.

MHC II epitopes of 15-mer length were predicted using three different tools, namely, IEDB combined server, TepiTool, and NetMHCIIpan. In TepiTool, MHC-II epitopes were searched using 'the panel of 27 most frequent A & B alleles', with the inclusion of HLA-DR, HLA-DP, and HLA-DQ alleles. In the 'prediction method', the default IEDB was applied, and a 'moderate number of peptides' was selected for the default setting of 'peptides' since the epitope length

was 15-mer. Finally, the peptides were predicted based on the IC50 cut-off value of less than or equal to 1000 nM. In NetMHCIIpan, 15-mer peptides were predicted for binding to the 27 MHC-II alleles from the reference list. The default threshold value for strong binders (% rank) was set as 1 and that for weak binders (% Rank) was set as 5. The results were sorted according to the prediction score; the strong binders were listed first and subjected to further analysis. All 9-mer and 15-mer peptides were predicted for their binding affinity to 27 MHC class I and MHC class II molecules, which accounted for 97% of HLA-A and HLA-B allelic variants in most ethnicities [52].

### Population coverage analysis of T-cell epitopes

The IEDB analysis resource tool was used to analyze the population coverage of predicted MHC class I and class II epitopes using the allele frequency database of 115 countries [53]. The population coverage was analysed by entering the following data: the number of epitopes, epitope, MHC-restricted alleles as predicted by the epitope prediction servers, and the populations in which the coverage is to be checked. The MHC-I alleles (n = 27) that bound to the predicted epitopes were checked using IEDB. For MHC-II alleles, twenty three alleles (n = 23) were available in the population coverage tool of IEDB (http://tools.iedb.org/population/) and four alleles (HLA-DRB3*01:01, HLA-DRB3*02:02, HLA-DRB4*01:01, and HLA-DRB5*01:01) were excluded from the analysis. The ability of the predicted T-cell epitopes to induce interferon-gamma (IFN-γ) response was assessed using the IFN epitope server. The IFN epitope tool uses an algorithm based on three models, motif-based, SVM-based, and hybrid approaches, for the prediction of IFN-γ-inducing epitopes [54].

### Analysis of antigenicity, allergenicity, and conservancy of predicted B- and T-cell epitopes

The antigenicity and allergenicity of epitopes were predicted using VaxiJen v2.0 and AllerTOP online servers, respectively. VaxiJen predicts protective antigens and subunit vaccines based on an alignment-independent method [55]. AllerTOP is a server for *in silico* prediction of allergens in a given antigen [56]. The epitope conservancy analysis tool from the IEDB analysis resource was used to compute the degree of the conservancy of an epitope within a given protein sequence [57].

### Immune simulation of multi-epitope vaccine

For designing multi-epitope vaccine, the predicted immunogenic epitopes were united with the help of linkers B-cell (GGGGS), HTL (KK), and CTL (AAY) [58]. To determine the immunogenicity and immune response profile of the multi-epitope vaccine consisting of the top predicted B- and T-cell epitopes, *in silico* immune simulation was conducted using the C-ImmSim server. The C-ImmSim server uses a position-specific scoring matrix and machine learning techniques to predict epitope and immune interactions [59]. The server simulates three components of the immune system found in mammals: (i) the bone marrow, where hematopoietic stem cells differentiate into cells of lymphoid and myeloid lineages; (ii) the thymus, where naive T-cells are selected to avoid autoimmunity; (iii) a tertiary lymphatic organ such as lymph nodes. All default simulation parameters were selected, with time steps set at 1, 84, and 168 (each time step is 8 hours, and time step 1 is injection at time = 0). Therefore, three injections were administered four weeks apart, without lipopolysaccharide (LPS). The Simpson index D (a measure of diversity) was interpreted from the plot.

## Results

### Prediction of B-cell epitopes of S, E, M, and N proteins of SARS-CoV-2

In this study, we predicted B- and T-cell epitopes using multiple prediction servers by leveraging existing immunological knowledge and the genetic closeness of SARS-CoV-2 with SARS-CoV [4–6]. Previous studies have identified epitopes mainly on the S protein of SARS-CoV-2 (Table 1) using a limited number of prediction servers [27–38]. Using well-established prediction tools, we identified potential B- and T-cell epitopes in the structural proteins (S, E, M, and N) of SARS-CoV-2 (Fig 1a). By selecting the top linear B-cell epitopes predicted using BepiPred, Bcepred, and ABCpred servers, we obtained 20 linear B-cell epitopes (11 for S, 6 for N, 2 for M, and 1 for E proteins) in the structural proteins of SARS-CoV-2 (Table 2). A heat map was generated using R software that showed the distribution of antigenic B-cell epitopes across the length of SARS-CoV-2 structural proteins (Fig 1b and 1c). Interestingly, some of the antigenic epitopes predicted by at least two servers are clearly delineated in the heat map.

In the absence of bioinformatics tools to analyze the common epitopes, we manually sorted the epitopes and found conserved epitopes of S (aa 407–416, aa421-427, aa1028-1049, aa1254-1273), N (aa173-189, aa235-247), M (aa163-182), and E (aa 58–68) proteins that are common to both SARS-CoV and SARS-CoV-2 (Table 3). The B-cell epitopes were mapped on the 3D structure of the structural proteins of SARS-CoV and SARS-CoV-2 and visualized using BIOVIA Discovery Studio 2017 R2. The images predicted the probable localization of epitopes on the surface of the 3D structure of S, E, M, and N proteins (Fig 2a–2e). In addition, the high alpha-helical content of most peptides, as predicted by the high Agadir scores, indicated the stability of peptides in solution [47].

**Table 1. A summary of literature review showing the bioinformatically predicted B- and T-cell epitopes identified in the structural proteins of SARS-CoV-2.**

| Sl No | Type of protein | Predicted B cell epitope | Predicted T cell epitope | Reference |
|---|---|---|---|---|
| 1 | Spike protein (S) | Yes | Yes | [20] |
| | 4 structural proteins and 5 non-structural proteins (open reading frame (ORF)3a, ORF6, ORF7a, ORF8, ORF10); and 1 non-structural polyprotein (ORF1ab) | - | Yes | |
| 2 | S | Yes | Yes | [21] |
| 3 | S | Yes | Yes | [22] |
| 4 | N | Yes | Yes | [23] |
| 5 | S | Yes | Yes | [24] |
| 6 | S | Yes | - | [25] |
| | S | - | Yes | |
| | M | | | |
| | N | | | |
| 7 | S | Yes | Yes | [27] |
| 8 | S | Yes | Yes | [28] |
| 9 | S | Yes | Yes | [29] |
| | N | | | |
| | M | | | |
| | Orf3a protein | | | |
| | Orf 1ab protein | | | |
| 10 | S, E, M | Yes | - | [30] |
| 11 | S | Yes | Yes | [31] |

The findings of the present *in silico* study have been provided, and the gap in the study has been addressed.

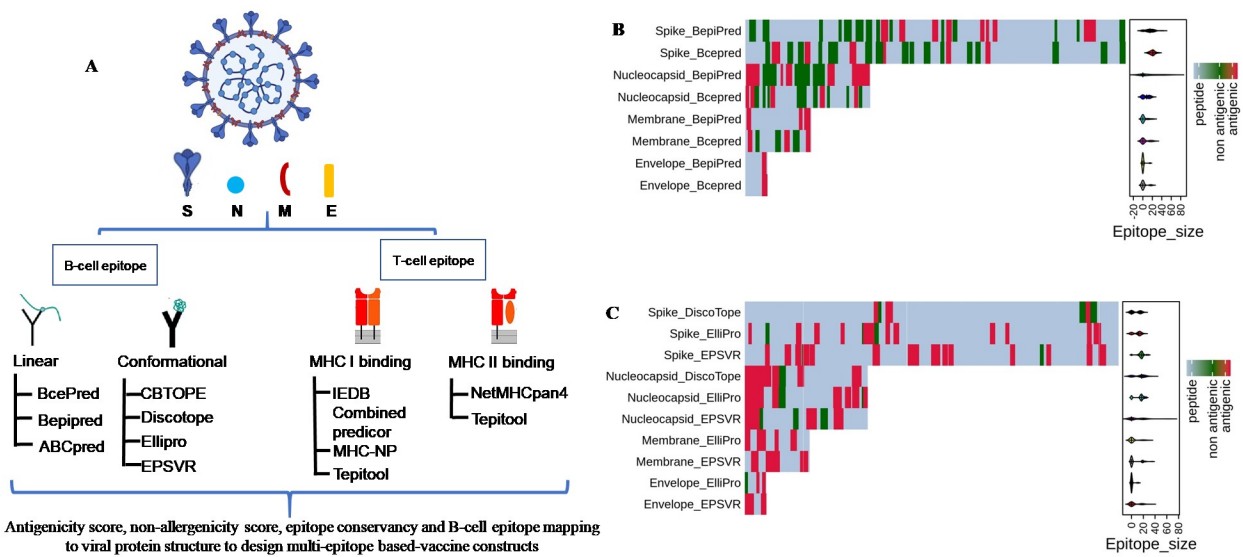

**Fig 1. (a)** Schematic representation of the *in-silico* workflow for the prediction of potential B- and T-cell epitopes of structural proteins (spike [S], envelope [E], membrane [M], and nucleocapsid [N] of SARS-CoV-2 and SARS-CoV. Summary of SARS-CoV-2-derived B-cell epitopes. Heat map showing the distribution of **(b)** linear (continuous) and **(c)** conformational (discontinuous) B-cell epitopes across the protein sequences of spike (1273 aa), nucleocapsid (419 aa), membrane (222 aa), and envelope (75 aa) proteins of SARS-CoV-2. Red represents the antigenic epitopes predicted using the methods described in Fig 1a.

The Cryo-EM structure of the S protein of SARS-CoV-2 (PDB: 6VSB) is available from residues 1 to 1208, but it lacks the 65 residues of the C-terminal region of the protein [3]. To overcome this problem, we used I-TASSER-modelled structures of the S protein of SARS-CoV-2 [60]. We compared I-TASSER-modelled structures with cryo-EM solved structures and observed high similarity, with a root mean square deviation (RMSD) of approximately 1.3 Å (SARS-CoV) and 2.0 Å (SARS-CoV-2). Similarly, the structures of E, M, and N proteins were modelled and used as query and input structures to identify potent conformational B-cell epitopes using CBTOPE, DiscoTope, ElliPro, and EPSVR servers. We filtered and selected the top conformational B-cell epitopes that were predicted by at least two prediction servers used in this study. We obtained a total of 21 conformational B-cell epitopes (9 for S, 8 for N, 3 for M, and 1 for E proteins) in the structural proteins of SARS-CoV-2 (Table 4). We observed conformational B-cell epitopes of S, E, M, and N proteins common to both SARS-CoV-2 and SARS-CoV with a high degree of epitope conservancy (Table 5). Furthermore, we noticed that the predicted conformational epitopes are likely to be localized on the accessible region of the 3D structure of SARS-CoV-2 proteins, as visualized by BIOVIA Discovery Studio 2017 R2 (Fig 3a–3e).

We performed sequence alignment of RBDs of the S protein and found that some of the linear and conformational B-cell epitopes are located in the RBD of the S protein of SARS-CoV-2 (Fig 4, inset Table). A previous study has identified four linear B-cell immunodominant (ID) sites on the RBD of the S protein located at aa330-349, aa375-394, aa450-469, and aa480-499 with an average positive rate of $\geq$ 50% among all 39 patients [61]. Although mice immunized with the entire RBD, aa370-395, and aa435-479 generated high titers of specific antibodies, these antibodies demonstrated weak neutralizing activity [61]. The identification of these epitopes in our present study clearly suggests that the bioinformatically predicted epitopes contribute to the immunogenicity of the S protein (Table 2). Interestingly, we observed that CR3022, a monoclonal antibody targeting a highly conserved cryptic epitope, has 20 out of 28 binding residues of the SARS-CoV-2 located in the RBD of the S protein (aa370-395; Table 2 & Fig 4) [10,

**Table 2. Linear B-cell epitopes of structural proteins of SARS-CoV-2.**

| SARS-CoV-2 protein | Amino acid position | Linear B-cell epitopes | Length | Server | Antigenicity by VaxiJen (T = 0.4) | Allergenicity by AllerTOP v. 2.0 | Agadir Score | Conservancy analysis by IEDB tool | | Experimentally verified epitopes / ViPR reference |
|---|---|---|---|---|---|---|---|---|---|---|
| | | | | | | | | SARS-CoV-2 | SARS-CoV | |
| Spike (S) | 177–196 | MDLEGKQGNFKNLREFVFKN | 20 | 2 | Antigen (0.6174) | Non-allergen | 1.32 | 99.43% (174/175) | 0.31% (1/326) | [74] |
| | 331–356 | NITNLCPFGEVFNATRFASVYAWNRK | 26 | 1 & 2 | Antigen (0.5525) | Allergen | 0.55 | 100.00% (175/175) | 0.00% (0/326) | [53, 36] |
| | 370–395 | NSASFSTFKCYGVSPTKLNDLCFTNV | 26 | 1 | Antigen (1.3609) | Non-allergen | 0.39 | 100.00% (175/175) | 0.00% (0/326) | [53][*In silico* predicted, 24, 25, 26] |
| | 403–427 | RGDEVRQIAPGQTGKIADYNYKLPD | 25 | 1 & 2 | Antigen (1.0356) | Non-allergen | 0.45 | 100.00% (175/175) | 0.00% (0/326) | [*In silico* predicted, 20, 22, 25, partial sequence from aa 405–418, 27] |
| | 437–461 | NSNNLDSKVGGNYNYLYRLFRKSNL | 25 | 1 & 2 | Antigen (0.4015) | Non-allergen | 4.42 | 100.00% (175/175) | 0.00% (0/326) | [10, 53, 69, 74] {[*In silico* predicted partial sequence from aa 440–450 [20], aa 441–448 [27]} |
| | 483–493 | VEGFNCYFPLQ | 11 | 1 | Antigen (0.5612) | Allergen | 0.16 | 100.00% (175/175) | 0.00% (0/326) | [10, 35] |
| | 525–546 | CGPKKSTNLVKNKCVNFNFNGL | 22 | 1 & 2 | Antigen (0.7688) | Allergen | 0.28 | 100.00% (175/175) | 0.31% (1/326) | [*In silico* predicted, 25] |
| | 653–666 | AEHVNNSYECDIPI | 14 | 1 & 2 | Antigen (0.6687) | Non-allergen | 0.03 | 99.43% (174/175) | 0.31% (1/326) | [71][*In silico* predicted partial sequence from aa 657–664, 29 and aa656-660, 30] |
| | 1028–1049 | KMSECVLGQSKRVDFCGKGYHL | 22 | 1 & 2 | Antigen (0.7717) | Allergen | 0.41 | 100.00% (175/175) | 84.05% (274/326) | [74] |
| | 1191–1211 | KNLNESLIDLQELGKYEQYIK | 21 | 2 | Antigen (0.5738) | Non-allergen | 3.24 | 98.29% (172/175) | 84.36% (275/326) | [74] |
| -Nucleocapsid (N) | 1253–1273 | CCKFDEDDSEPVLKGVKLHYT | 21 | 1 & 2 | Antigen (0.9101) | Non-allergen | 0.26 | 94.86% (166/175) | 83.74% (273/326) | [55]IEDB ID: 6476 [59][*In silico* predicted partial sequence from aa 1256–1265, 22] |
| | 2–13 | SDNGPQNQRNAP | 12 | 1 & 2 | Antigen (0.6672) | Non-allergen | 0.23 | 100.00% (185/185) | 0.00% (0/301) | - |
| | 32–46 | RSGARSKQRRPQGLP | 15 | 1, 2 & 3 | Antigen (0.8571) | Non-allergen | 0.37 | 99.46% (184/185) | 0.00% (0/301) | - |

(*Continued*)

**Table 2.** (Continued)

| SARS-CoV-2 protein | Amino acid position | Linear B-cell epitopes | Length | Server | Antigenicity by VaxiJen (T = 0.4) | Allergeicity by AllerTOP v. 2.0 | Agadir Score | Conservancy analysis by IEDB tool | | Experimentally verified epitopes / ViPR reference |
|---|---|---|---|---|---|---|---|---|---|---|
| | | | | | | | | SARS-CoV-2 | SARS-CoV | |
| | 173–190 | AEGSRGGSQASSRSSSRS | 18 | 1, 2 & 3 | Antigen (0.8081) | Non-allergen | 0.21 | 99.46% (184/185) | 79.73% (240/301) | IEDB ID: 22481 [59][In silico predicted, 23] |
| | 227–266 | LNQLESKMSGKGQQQGQTVTKKSAAEASKKPRQKRTATK | 40 | 1, 2 & 3 | Antigen (0.5387) | Non-allergen | 1.9 | 100.00% (185/185) | 0.00% (0/301) | [In silico predicted, partial sequence from aa 235–243, aa 249–263, 23] |
| | 274–283 | FGRRGPEQTQ | 10 | 2 | Antigen (1.0355) | Allergen | 0 | 100.00% (185/185) | 83.39% (251/301) | [73] |
| | 366–394 | TEPKKDKKKKADETQALPQRQKKQQTVTL | 29 | 1, 2 & 3 | Antigen (0.5248) | Non-allergen | 0.63 | 100.00% (185/185) | 0.00% (0/301) | {[In silico predicted, partial sequence from aa 363–379 [23] and aa 355–394 [29]} |
| Membrane (M) | 101–119 | RLFARTRSMWSFNPETNIL | 19 | 2 | Antigen (0.4166) | Allergen | 0.17 | 98.82% (168/170) | 82.46% (221/268) | [73] |
| | 160–182 | DIKDLPKEITVATSRTLSYYKLG | 23 | 2 & 3 | Antigen (0.7442) | Non-allergen | 0.31 | 98.24% (167/170) | 79.48% (213/268) | - |
| Envelope (E) | 57–75 | YVYSRVKNLNSSRVPDLLV | 19 | 1, 2 & 3 | Antigen (0.565) | Non-allergen | 0.2 | 100.00% (177/177) | 0.00% (0/245) | [In silico predicted partial sequence from aa 656–660, 30] |

Four structural proteins of SARS-CoV-2, namely spike (S), envelope (E), membrane (M,) and nucleocapsid (N) proteins, were subjected to prediction of B-cell epitopes. List of B-cell epitopes predicted by at least two servers, with the corresponding prediction servers numbered as (1) for BepiPred, (2) for Bcepred, and (3) for ABCpred.

**Table 3. Linear B-cell epitopes common to SARS-CoV-2 and SARS-CoV.**

| CoVs protein | Amino acid position | | Common linear B-cell epitopes | Length | Probable Antigenicity by VexiJen (T = 0.4) | Allergeicity by AllerTope v. 2.0 | Agadir score | Conservancy analysis by IEDB tool | |
|---|---|---|---|---|---|---|---|---|---|
| | SARS-CoV-2 | SARS-CoV | | | | | | SARS-CoV-2 | SARS-CoV |
| Spike (S) | 281–291 | 268–278 | ENGTITDAVDC | 11 | Antigen (0.5860) | Allergen | 0.09 | 100.00% (175/175) | 85.58% (279/326) |
| | 374–383 | 361–370 | FSTFKCYGVS | 10 | Antigen (0.7239) | Allergen | 0.05 | 100.00% (175/175) | 86.20% (281/326) |
| | 407–416 | 394–403 | VRQIAPGQTG | 10 | Antigen (1.3856) | Non-allergen | 0 | 100.00% (175/175) | 85.89% (280/326) |
| | 421–427 | 408–414 | YNYKLPD | 7 | Antigen (1.1454) | Allergen | 0 | 100.00% (175/175) | 86.20% (281/326) |
| | 694–701 | 676–683 | AYTMSLGA | 8 | Antigen (0.6924) | Non-allergen | 0.02 | 100.00% (175/175) | 85.58% (279/326) |
| | 828–833 | 810–815 | LADAGF | 6 | Antigen (1.1934) | Non-allergen | 0.01 | 100.00% (175/175) | 85.58% (279/326) |
| | 1028–1049 | 1010–1031 | KMSECVLGQSKRVDFCGKGYHL | 22 | Antigen (0.7717) | Allergen | 0.41 | 100.00% (175/175) | 84.05% (274/326) |
| | 1112–1118 | 1094–1100 | PQIITTD | 7 | Antigen (0.6387) | Allergen | 0 | 100.00% (175/175) | 85.28% (278/326) |
| | 1157–1165 | 1139–1147 | KNHTSPDVD | 9 | Antigen (0.7809) | Allergen | 0 | 100.00% (175/175) | 85.89% (280/326) |
| | 1254–1273 | 1236–1255 | CKFDEDDSEPVLKGVKLHYT | 20 | Antigen (0.8247) | Non-allergen | 0.28 | 94.86% (166/175) | 83.74% (273/326) |
| Nucleocapsid (N) | 38–46 | 39–47 | KQRRPQGLP | 9 | Antigen (0.9734) | Non-allergen | 0 | 99.46% (184/185) | 85.05% (256/301) |
| | 121–126 | 122–127 | LPYGAN | 6 | Antigen (0.7670) | Non-allergen | 0 | 100.00% (185/185) | 85.38% (257/301) |
| | 173–189 | 174–190 | AEGSRGGSQASSRSSSR | 17 | Antigen (0.7412) | Non-allergen | 0.2 | 99.46% (184/185) | 79.73% (240/301) |
| | 228–233 | 229–234 | NQLESK | 6 | Antigen (0.5533) | Non-allergen | 0.03 | 100.00% (185/185) | 84.72% (255/301) |
| | 235–247 | 236–248 | SGKGQQQQGQTVT | 13 | Antigen (0.5562) | Non-allergen | 0.15 | 100.00% (185/185) | 79.73% (240/301) |
| | 339–344 | 340–345 | LDDKDP | 6 | Antigen (1.9529) | Non-allergen | 0 | 98.38% (182/185) | 85.71% (258/301) |
| | 382–389 | 383–390 | LPQRQKKQ | 8 | Antigen (1.3573) | Non-allergen | 0.13 | 100.00% (185/185) | 82.72% (249/301) |
| Membrane (M) | 41–48 | 40–47 | NRNRFLYI | 8 | Antigen (0.6221) | Non-allergen | 0.04 | 98.82% (168/170) | 80.22% (215/268) |
| | 136–144 | 135–143 | SELVIGAVI | 9 | Antigen (0.6409) | Non-allergen | 0.05 | 98.82% (168/170) | 82.46% (221/268) |
| | 163–182 | 162–181 | DLPKEITVATSRTLSYYKLG | 20 | Antigen (0.4328) | Non-allergen | 0.4 | 98.24% (167/170) | 79.48% (213/268) |
| Envelope (E) | 58–68 | 58–68 | VYSRVKNLNSS | 11 | Antigen (0.7845) | Non-allergen | 0.18 | 100.00% (177/177) | 83.27% (204/245) |

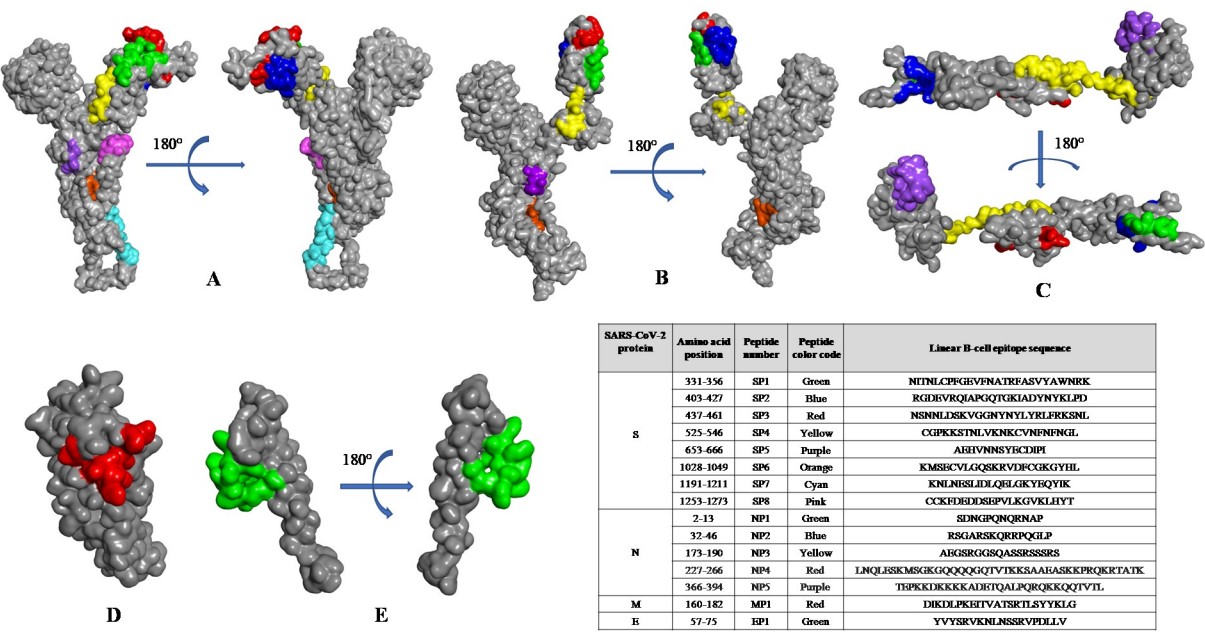

**Fig 2. Mapping of linear B-cell epitopes on the three-dimensional structure of SARS-CoV-2 structural proteins.** Localization of top predicted monomeric B-cell epitopes (inset table) on the **(a)** modeled structure and **(b)** crystal structure (PDB: 6VSB) of the spike protein, **(c)** nucleocapsid, **(d)** membrane, and **(e)** envelope proteins, as observed using BIOVIA Discovery Studio 2017 R2.

62]. The B38, which is a SARS-CoV-2-specific human neutralizing monoclonal antibody, has 9 out of 18 binding residues on the RBD of SARS-CoV-2 [11, 12, 62]. B38 is involved in both B38 and hACE2 interaction located in the ID of predicted linear and conformational B-cell epitopes (aa403-427, aa437-461, and aa483-493; Tables 2 and 4). Similarly, another monoclonal antibody (S309) known to neutralize both SARS-CoV-2 and SARS-CoV, has 11 out of 21 binding residues in ID sites [10, 62]. A previous computational study has predicted aa656-660 of the S protein and aa60-65 of the E protein as dominant conformational B-cell epitopes [37], and it is worthy to note that the same epitopes have been predicted in our study (Table 4). A recent study has predicted that conformational epitopes are mainly localized at aa405-427 and aa439-505 regions of the tertiary structure of the S protein [38]. The highly conserved linear B-cell epitope (aa1253-1273, Table 2) predicted in the S protein has been shown to elicit IgG antibodies in some patients with COVID-19 when tested using synthetic peptides spanning aa1256-1273 [63]. A previous study using pools of overlapping linear B-cell peptides has found IgG immunodominant regions on the S protein of SARS-CoV-2. These regions are recognized by convalescent sera from patients with COVID-19, and the region aa562-580 is localized near the RBD of the S protein [64]. The partial sequence of this immunogenic peptide is computationally predicted as a conformational B-cell epitope in our study (Table 4). Findings of computational prediction of epitopes reported in this study, as well as the published experimental observations, suggest the association of linear ID B-cell epitopes with conformational epitopes. However, further studies are needed to identify the functions of the predicted linear and conformational epitopes for the rational design of *de novo* peptide-based vaccines.

## Prediction of MHC I-binding epitopes of S, E, M, and N proteins of SARS-CoV-2

A few studies on SARS-CoV-2-specific T-cell responses and their role in protective immunity [65] found that recovered patients with COVID-19 show CD4+ and CD8+ memory responses

**Table 4. Conformational B-cell epitopes of structural proteins of SARS-CoV-2.**

| SARS-CoV-2 protein | Sl. No. | Conformational B-cell epitopes | Length | Server | Probable Antigenicity by VaxiJen (T = 0.4) | Allergicity by AllerTOP v. 2.0 | Agadir Score | Conservancy analysis by IEDB tool | | Experimentally verified epitopes / ViPR reference |
|---|---|---|---|---|---|---|---|---|---|---|
| | | | | | | | | SARS-CoV-2 | SARS-CoV | |
| Spike (S) | 1 | S1161,P1162,D1163,V1164,D1165,L1166,G1167,D1168,I1169,S1170,G1171,I1172,N1173,A1174,S1175,V1176,V1177,N1178 | 18 | 1,2,3,4 | Antigen (0.872) | Allergen | 0.29 | 100.00% (175/175) | 83.44% (272/326) | - |
| | 2 | Y904,R905,F906,N907,G908,I909,G910,V911,T912,Q913,N914,V915,L916,Y917,E918,N919,Q920,K921 | 18 | 1,4 | Antigen (0.8334) | Non-allergen | 0.21 | 100.00% (175/175) | 85.89% (280/326) | - |
| | 3 | T887,F888,G889,A890,G891,A892,A893,I894,Q895,I896,P897,F898,A899,M900 | 14 | 1,3 | Antigen (0.5837) | Non-allergen | 0.33 | 100.00% (175/175) | 85.58% (279/326) | - |
| | 4 | Q1201,E1202,L1203,G1204,K1205,Y1206,E1207,Q1208,Y1209,I1210,K1211,W1212 | 12 | 3,4 | Antigen (0.5743) | Allergen | 0.42 | 94.86% (166/175) | 82.52% (269/326) | - |
| | 5 | T1027,K1028,S1030,E1031,L1034,G1035,S1037,K1038,R1039,V1040,D1041,F1042 | 12 | 1,4 | Antigen (1.6883) | Non-allergen | 0.15 | 100.00% (175/175) | 84.05% (274/326) | - |
| | 6 | R646,A647,G648,L650,I651,E654,H655,V656,N657,N658,S659,Y660,E661,C662,D663,I664,P665 | 17 | 1,4 | Antigen (0.5886) | Non-allergen | 0.16 | 99.43% (174/175) | 0.31% (1/326) | [In silico predicted, 30] |
| | 7 | F175,L176,M177,D178,L179,E180,G181,K182,Q183,G184,N185,F186,K187,N188,L189,R190 | 16 | 1,3,4 | Antigen (0.8797) | Non-allergen | 0.35 | 99.43% (174/175) | 0.31% (1/326) | - |
| | 8 | I472,Y473,Q474,A475,G476,S477,T478,P479,C480,N481,G482,V483,E484,G485,F486,N487,C488,Y489,F490,P491,L492 | 21 | 1 | Non-antigen (0.2378) | Allergen | 0.16 | 98.86% (173/175) | 0.00% (0/326) | [10] |
| | 9 | P491,L492,Q493,S494,Y495,G496,F497,Q498,P499,T500,V501 | 10 | 2 & 3 | Antigen (0.5224) | Allergen | 0.01 | 100.00% (175/175) | 0.00% (0/326) | [10] |
| Nucleocapsid (N) | 1 | D22,S23,T24,G25,S26,N27,Q28,N29,G30,E31,R32,S33,G34,A35,R36,S37,K38,Q39,R40 | 19 | 2,3,4 | Antigen (0.6738) | Non-allergen | 0.27 | 100.00% (185/185) | 0.00% (0/301) | - |
| | 2 | R95,G96,G97,D98,G99,R100,M101,K102,D103,L104,S105,P106,R107,W108,Y109,F110,Y111,Y112,L113,G114 | 20 | 1,2,3 | Antigen (1.1156) | Non-allergen | 0.6 | 100.00% (185/185) | 0.00% (0/301) | - |
| | 3 | P365,T366,E367,P368,R369,K370,D371,K372,K373,K374,K375,A376,D377,E378,T379,Q380,A381,L382,P383,Q384,R385,Q386,K387 | 23 | 1,2,3 | Antigen (0.4426) | Non-allergen | 0.69 | 100.00% (185/185) | 0.00% (0/301) | - |
| | 4 | A267,Y268,N269,V270,T271,Q272,A273,F274,G275,R276,R277,G278,P279,E280,Q281,T282,Q283,G284,N285,F286,G287,D288,Q289,E290,L291,I292,R293,Q294,G295,T296,D297,Q298,K299,H300 | 34 | 1,4 | Antigen (0.6367) | Non-allergen | 0.71 | 100.00% (185/185) | 0.33% (1/301) | - |
| | 5 | E378,T379,Q380,A381,L382,P383,Q384,R385,Q386,K387,K388,Q389,Q390,T391,V392,T393,L394,L395,P396 | 19 | 1,2 | Antigen (0.7896) | Non-allergen | 0.34 | 100.00% (185/185) | 0.00% (0/301) | - |
| | 6 | P302,Q303,I304,A305,Q306,F307,A308,P309,S310,F315,G316,R319,I320,G321,E323,V324,T325,P326,S327,G328,W330 | 21 | 1,3,4 | Antigen (0.809) | Non-allergen | 0.04 | 100.00% (185/185) | 83.06% (250/301) | - |
| | 7 | L104,S105,P106,R107,W108,Y109,F110,Y111,Y112,L113,G114,T115,G116,P117 | 14 | 2,3 | Antigen (1.3764) | Non-allergen | 0.77 | 100.00% (185/185) | 83.39% (251/301) | - |
| | 8 | Q272,A273,F274,G275,R276,R277,G278,P279,E280,Q281,T282,Q283,G284,N285,F286,G287,D288,Q289 | 18 | 1,4 | Antigen (0.8885) | Allergen | 0.1 | 100.00% (185/185) | 85.71% (258/301) | - |

(*Continued*)

Table 4. (Continued)

| SARS-CoV-2 protein | Sl. No. | Conformational B-cell epitopes | Length | Server | Probable Antigenicity by VaxiJen (T = 0.4) | Allergeicity by AllerTOP v. 2.0 | Agadir Score | Conservancy analysis by IEDB tool | | Experimentally verified epitopes / ViPR reference |
|---|---|---|---|---|---|---|---|---|---|---|
| | | | | | | | | SARS-CoV-2 | SARS-CoV | |
| Membrane (M) | 1 | L156,G157,R158,C159,D160,I161,K162,D163, L164,P165,A171,T172,S173,R174,T175,L176, S177 | 17 | 1,3 | Antigen (0.8363) | Non-allergen | 0.09 | 98.24% (167/ 170) | 80.97% (217/268) | - |
| | 2 | G189,D190,F193,A194,A195,S197,Y199,R200, I201,G202,N203,Y204,K205,L206,N207,T208, D209,H210 | 18 | 1,4 | Antigen (0.4423) | Allergen | 0.72 | 98.82% (168/ 170) | 0.75% (2/ 268) | - |
| | 3 | L176,S177,Y178,Y179,K180,L1801G182,A183, S184,Q185,R186,V187 | 12 | 1, 3, 4 | Antigen (0.6421) | Allergen | 0.11 | 98.82% (168/ 170) | 82.46% (221/268) | - |
| Envelope (E) | 1 | S60,R61,V62,K63,N64,L65,N66,S67,S68,R69, V70,P71,D72,L73,L74 | 15 | 1,3,4 | Antigen (0.7404) | Allergen | 0.12 | 100.00% (177/177) | 0.00% (0/ 245) | [In silico predicted, 30] |

The list of conformational B-cell epitopes predicted by at least two servers, with the corresponding prediction servers numbered as (1) for CBTOPE, (2) for DiscoTope, (3) for ElliPro, and (4) for EPSVR.

**Table 5. Conformational B-cell epitopes common to SARS-CoV-2 and SARS-CoV.**

| Protein, PDB/ I-TASSER | Conformational B-cell epitopes | | Length | Server | | Probable Antigenicity by VexiJen (T = 0.4) | Allergeicity by AllerTope v. 2.0 | Agadir Score | Conservancy analysis by IEDB tool | |
|---|---|---|---|---|---|---|---|---|---|---|
| | SARS-CoV-2 | SARS-CoV | | SARS-CoV-2 | SARS-CoV | | | | SARS-CoV-2 | SARS-CoV |
| Spike (S) | S1161,P1162, D1163,V1164, D1165,L1166, G1167,D1168, I1169,S1170, G1171,I1172, N1173,A1174, S1175,V1176, V1177,N1178 | S1143,P1144, D1145,V1146, D1147,L1148, G1149,D1150, I1151,S1152, G1153,I1154, N1155,A1156, S1157,V1158, V1159,N1160 | 18 | 1,2,3,4 | 1,2,3,4 | Antigen (0.872) | Allergen | 0.29 | 100.00% (175/175) | 83.44% (272/326) |
| | Q1201,E1202, L1203,G1204, K1205,Y1206, E1207,Q1208, Y1209,I1210, K1211,W1212 | Q1183,E1184, L1185,G1186, K1187,Y1188, E1189,Q1190, Y1191,I1192, K1193,W1194 | 12 | 3,4 | 4 | Antigen (0.5743) | Allergen | 0.42 | 94.86% (166/175) | 82.52% (269/326) |
| | C1254,K1255, F1256,D1257, E1258,D1259, D1260,S1261, E1262,P1263, V1264,L1265, K1266,G1267, V1268,K1269, L1270,H1271 | C1236,K1237, F1238,D1239, E1240,D1241, D1242,S1243, E1244,P1245, V1246,L1247, K1248,G1249, V1250,K1251, L1252,H1253 | 18 | 1 | 1 | Antigen (0.5883) | Non-allergen | 0.24 | 94.86% (166/175) | 83.74% (273/326) |
| Spike (S) SARS-CoV-2 (6VSB) & SARS-CoV (5WRG) | F797,G798,G799, F800,N801,F802, S803,Q804,I805 | F779,G780,G781, F782,N783,F784, S785,Q786,I787 | 9 | 3 | 3 | Antigen (1.2730) | Non-allergen | 0.01 | 99.43% (174/175) | 83.44% (272/326) |
| | Q901,M902,A903, Y904,R905,F906, N907,G908,I909 | Q883,M884,A885, Y886,R887,F888, N889,G890,I891 | 9 | 3 | 3 | Antigen (0.6803) | Non-allergen | 0.04 | 100.00% (175/175) | 86.81% (283/326) |
| | G1035,Q1036, S1037,K1038, R1039,V1040, D1041,F1042 | G1017,Q1018, S1019,K1020, R1021,V1022, D1023,F1024 | 8 | 1 | 1 & 3 | Antigen (1.9298) | Allergen | 0.03 | 100.00% (175/175) | 84.05% (274/326) |
| Nucleocapsid (N) | L104,S105,P106, R107,W108,Y109, F110,Y111,Y112, L113,G114,T115, G116,P117 | L105,S106,P107, R108,W109,Y110, F111,Y112,Y113, L114,G115,T116, G117,P118 | 14 | 2,3 | 1,2 | Antigen (1.3764) | Non-allergen | 0.77 | 100.00% (185/185) | 83.39% (251/301) |
| | Q272,A273,F274, G275,R276,R277, G278,P279,E280, Q281,T282,Q283, G284,N285,F286, G287,D288,Q289 | Q273,A274,F275, G276,R277,R278, G279,P280,E281, Q282,T283,Q284, G285,N286,F287, G288,D289,Q290 | 18 | 1,4 | 1,2,4 | Antigen (0.8885) | Allergen | 0.1 | 100.00% (185/185) | 85.71% (258/301) |
| Membrane (M) | L176,S177,Y178, Y179,K180, L1801G182,A183, S184,Q185,R186, V187 | L175,S176,Y177, Y178,K179,L180, G181,A182,S183, Q184,R185,V186 | 12 | 1 | 1,3,4 | Antigen (0.6421) | Allergen | 0.11 | 98.82% (168/170) | 82.46% (221/268) |
| Envelope (E) | C43,C44,N45,I46, V47,N48,V49,S50 | C43,C44,N45,I46, V47,N48,V49,S50 | 9 | 3 | 3 | Antigen (0.8522) | Allergen | 0.02 | 100.00% (177/177) | 84.49% (207/245) |
| | S60,R61,V62,K63, N64,L65,N66,S67, S68 | S60,R61,V62,K63, N64,L65,N66,S67, S68 | 9 | 1,3,4 | 1,3,4 | Antigen (0.7404) | Allergen | 0.12 | 100.00% (177/177) | 0.00% (0/ 245) |

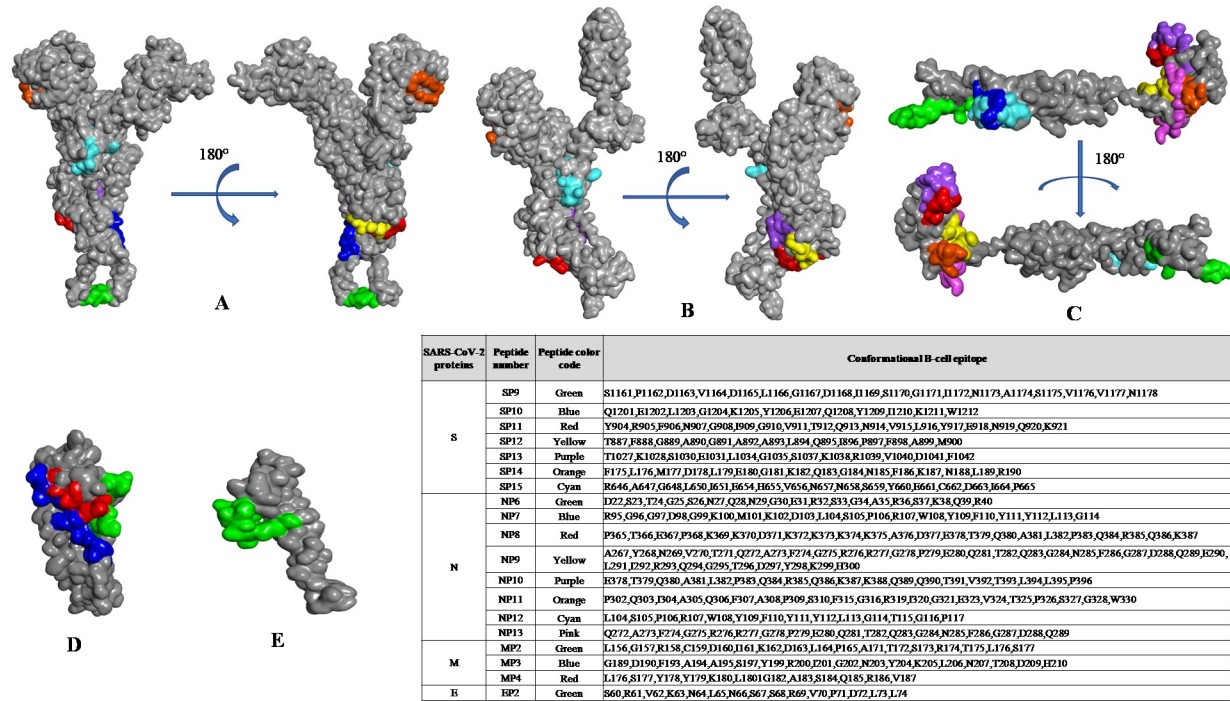

| SARS-CoV-2 proteins | Peptide number | Peptide color code | Conformational B-cell epitope |
|---|---|---|---|
| S | SP9 | Green | S1161,P1162,D1163,V1164,D1165,L1166,G1167,D1168,I1169,S1170,G1171,I1172,N1173,A1174,S1175,V1176,V1177,N1178 |
| | SP10 | Blue | Q1201,E1202,L1203,G1204,K1205,Y1206,E1207,Q1208,Y1209,I1210,K1211,W1212 |
| | SP11 | Red | Y904,R905,I906,N907,G908,I909,G910,V911,T912,Q913,N914,V915,L916,Y917,E918,N919,Q920,K921 |
| | SP12 | Yellow | T887,F888,G889,A890,G891,A892,A893,L894,Q895,I896,P897,F898,A899,M900 |
| | SP13 | Purple | T1027,K1028,S1030,E1031,L1034,G1035,S1037,K1038,R1039,V1040,D1041,F1042 |
| | SP14 | Orange | F175,L176,M177,D178,L179,I180,G181,K182,Q183,G184,N185,F186,K187,N188,L189,R190 |
| | SP15 | Cyan | R646,A647,G648,L650,I651,E654,H655,V656,N657,N658,S659,Y660,I661,C662,D663,I664,P665 |
| N | NP6 | Green | D22,S23,T24,G25,S26,N27,Q28,N29,G30,I31,R32,S33,G34,A35,R36,S37,K38,Q39,R40 |
| | NP7 | Blue | R95,G96,Q97,D98,Q99,K100,M101,K102,D103,L104,S105,P106,R107,W108,Y109,F110,Y111,Y112,L113,G114 |
| | NP8 | Red | P365,T366,E367,P368,K369,K370,D371,K372,K373,K374,K375,A376,D377,E378,T379,Q380,A381,L382,P383,Q384,R385,Q386,K387 |
| | NP9 | Yellow | A267,Y268,N269,V270,T271,Q272,A273,F274,G275,R276,R277,G278,P279,E280,Q281,T282,Q283,Q284,N285,F286,G287,D288,Q289,E290,L291,I292,R293,Q294,G295,T296,D297,Y298,K299,H300 |
| | NP10 | Purple | E378,T379,Q380,A381,L382,P383,Q384,R385,Q386,K387,K388,Q389,Q390,T391,V392,T393,L394,L395,P396 |
| | NP11 | Orange | P302,Q303,I304,A305,Q306,F307,A308,P309,S310,F315,G316,R319,I320,G321,I323,V324,T325,P326,S327,G328,W330 |
| | NP12 | Cyan | L104,S105,P106,R107,W108,Y109,F110,Y111,Y112,L113,G114,T115,G116,P117 |
| | NP13 | Pink | Q272,A273,F274,G275,R276,R277,G278,P279,E280,Q281,T282,Q283,Q284,N285,F286,G287,D288,Q289 |
| M | MP2 | Green | L156,G157,R158,C159,D160,I161,K162,D163,L164,P165,A171,T172,S173,R174,T175,L176,S177 |
| | MP3 | Blue | G189,D190,P193,A194,A195,S197,Y199,R200,I201,G202,N203,Y204,K205,L206,N207,T208,D209,H210 |
| | MP4 | Red | L176,S177,Y178,Y179,K180,L180,G182,A183,S184,Q185,R186,V187 |
| E | EP2 | Green | S60,R61,V62,K63,N64,L65,N66,S67,S68,R69,V70,P71,D72,L73,L74 |

**Fig 3. Mapping of conformational B-cell epitopes on the three-dimensional structure of SARS-CoV-2 structural proteins.** The selected B-cell epitopes are listed in the inset table, and the corresponding color shows the localization of predicted epitopes on the **(a)** modeled structure and **(b)** crystal structure (PDB: 6VSB) of the spike, **(c)** nucleocapsid, **(d)** membrane, and **(e)** envelope proteins, as observed using BIOVIA Discovery Studio 2017 R2.

to SARS-CoV-2 [16]. SARS-CoV-2-specific CD4+ T-cell responses were also frequently observed in unexposed healthy participants, suggesting the possibility of pre-existing cross-reactive immune memory to seasonal human coronaviruses [18, 66]. The access to information on SARS-CoV-2 proteins, and epitopes recognized by human T-cells, can greatly assist researchers in selecting potential epitopes, or target proteins for the future design of candidate vaccines.

We selected the high affinity-ranked top 2% peptides and found 55 non-overlapping peptides (26 CD8+ T-cell epitopes for S, 16 for N, 10 for M, and 3 for E proteins), as strong binders of MHC I molecule, based on the prediction of epitopes by at least two servers (Table 6). These strong MHC I binding peptides were assessed for their predicted capacity to elicit IFN-γ responses and are concurrently predicted to possess antigenic and non-allergenic properties. Notably, we predicted 16 CD8+ T-cell epitopes (5 CD8+ T-cell epitopes each for S and N proteins and 3 each for M and E proteins), which are common to structural proteins of both SARS-CoV-2 and SARS-CoV (Table 7).

## Prediction of MHC II-binding epitopes of S, E, M, and N proteins of SARS-CoV-2

Helper T cells, which are required for adaptive immune responses, help activate B cells to secrete antibodies and activate cytotoxic T cells to kill infected target cells. TepiTool [48] and NetMHCIIpan [51] were used to predict and identify high-affinity MHC II-binding epitopes based on 27 HLA class II molecules (HLA-DP, HLA-DQ, and HLA-DR). We found 25 non-overlapping CD4+ T-cell epitopes (15 CD4+ T-cell epitopes for S, 6 for N, 3 for M, and 1 for E

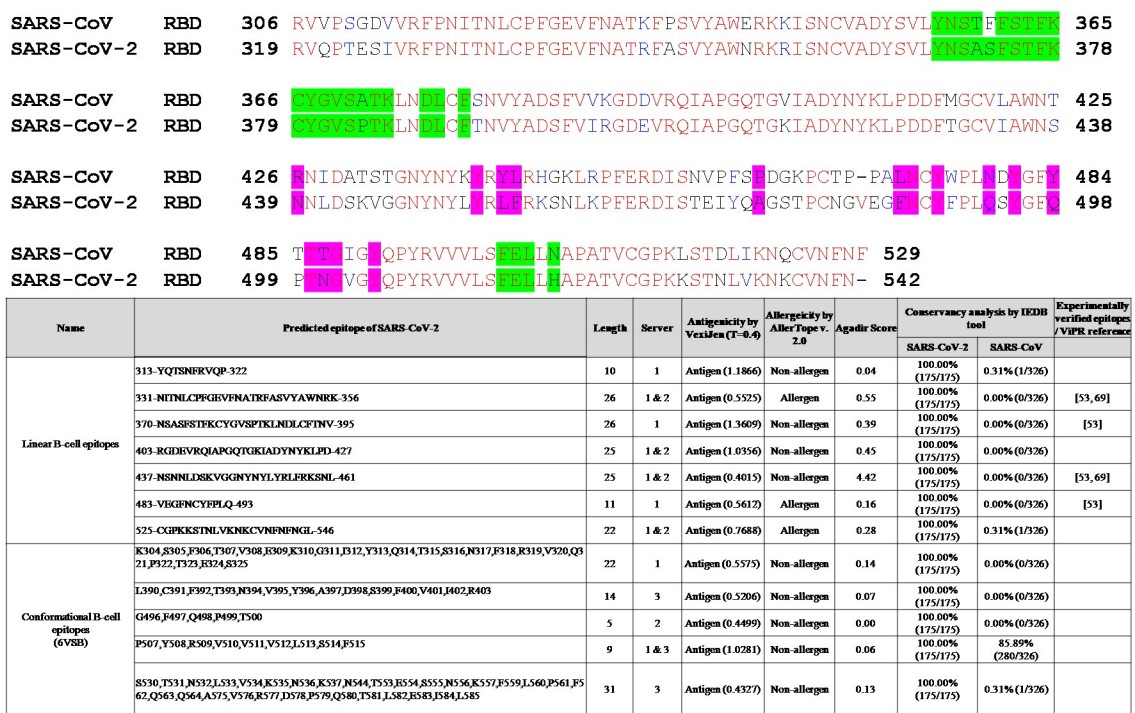

**Fig 4. Sequence alignment of receptor-binding domains (RBDs) of SARS-CoV-2 and SARS-CoV spike (S) proteins.** GenBank accession numbers of the S protein of SARS-CoV-2 and SARS-CoV are QHR63250.2 (SARS-CoV-2) and AAP30030.1 (SARS-CoV BJ01), respectively. ACE2-binding residues are colored magenta, and CR3022 epitope residues are colored bright green. The alignment was performed using Clustal Omega. The inset table shows the antigenic linear and conformational B-cell epitopes predicted in the RBD of the S protein of SARS-CoV-2.

proteins), which were predicted as antigenic and non-allergenic, as well as IFN-gamma-inducing property, with a high degree of conservancy (Table 8). Furthermore, we found 4 CD4+ T-cell epitopes for S, 5 for N, and 2 each for M and E proteins, which are common to both SARS-CoV-2 and SARS-CoV, with epitope conservancy in the range of 80 to 100% (Table 9). The allele-wise distribution of predicted MHC I and MHC II epitopes for the S, E, M, and N proteins of SARS-CoV-2 are presented in a heat map (Fig 5a and 5b).

On careful observation of computationally predicted epitopes, we found that the regions aa403-427 and aa437-461 of the S protein were predicted as both linear B-cell and MHC I-binding epitopes (Table 10). Similarly, the conserved C-terminal epitope of S (aa1253-1273) and E (aa57-75) proteins common to SARS-CoV and SARS-CoV-2 were predicted as both linear and conformational B-cell epitopes. Notably, the peptide aa366-394 in the N protein and the peptide aa160-182 in the M protein were predicted to induce both humoral and cell-mediated immune responses (Table 11).

The findings of our computational predictions in conjunction with previous experimental studies suggest that T-cell-based immunity might be generated largely against the S and N proteins of SARS-CoV-2; therefore, the S and N proteins of SARS-CoV-2 could be selected as target candidates for recombinant protein-based vaccines [14–16, 61–63]. CD4+ T-cell responses were observed predominately in the S protein, and the robustness of the T-cell response was correlated with the values of IgG and IgA titers of SARS-CoV-2. Among the structural proteins, the S and M proteins were mainly recognized as possible targets for CD8+ T cells of SARS-CoV-2 [16, 17, 65, 66].

**Table 6. List of predicted SARS-CoV-2-derived T-cell epitopes with high-affinity binding to MHC I molecules.**

| SARS-CoV-2 protein | Sl. No. | Peptide | Position | Alleles | Number of Alleles | Servers | Probable Antigenicity by Vexilen (Threshold = 0.4) | Probable Allergenicity by AllerTop | Conservancy analysis by IEDB tool | Conservancy analysis by IEDB tool SARS-CoV-2 | SARS-CoV | IFN gamma epitope | Experimentally verified epitopes / ViPR reference |
|---|---|---|---|---|---|---|---|---|---|---|---|---|---|
| Spike (S) | 1 | FSNVTWFHA | 59–67 | HLA-A*02:06, HLA-A*68:02 | 2 | 1 & 3 | Antigen (0.8156) | Non-allergen | 100.00% (175/175) | 0.31% (1/326) | Positive | IEDB ID: 1074888 [59] | 1 |
| | 2 | VTWFHAIHV | 62–70 | HLA-A*02:06, HLA-A*68:02, HLA-A*02:01, HLA-A*32:01, HLA-A*02:03 | 5 | 1, 2 & 3 | Antigen (0.5426) | Non-allergen | 100.00% (175/175) | 0.31% (1/326) | Positive | - | 2 |
| | 3 | GVYFASTEK | 89–97 | HLA-A*11:01, HLA-A*03:01, HLA-A*68:01, HLA-A*30:01 | 4 | 1, 2 & 3 | Antigen (0.7112) | Non-allergen | 100.00% (175/175) | 0.31% (1/326) | Positive | IEDB ID: 1075093 [59] [In silico predicted, 24, 26] | 3 |
| | 4 | TFEYVSQPF | 167–175 | HLA-A*23:01, HLA-A*24:02 | 2 | 1, 2 & 3 | Antigen (0.6641) | Non-allergen | 100.00% (175/175) | 0.31% (1/326) | Positive | IEDB ID: 1074869 [59] [In silico predicted, 22, 29] | 4 |
| | 5 | WTAGAAAYY | 258–266 | HLA-A*26:01, HLA-A*68:01, HLA-A*01:01, HLA-A*30:02, HLA-B*35:01, HLA-B*15:01, HLA-B*58:01 | 7 | 1, 2 & 3 | Antigen (0.6306) | Non-allergen | 100.00% (175/175) | 0.31% (1/326) | Positive | IEDB ID: 1075117 [59] [In silico predicted, 21, 22, 26] | 5 |
| | 6 | GAAAYYVGY | 261–269 | HLA-A*30:02, HLA-B*15:01, HLA-B*35:01, HLA-A*26:01, HLA-B*58:01, HLA-A*11:01, HLA-A*01:01 | 7 | 1, 2 & 3 | Antigen (0.6604) | Non-allergen | 100.00% (175/175) | 0.31% (1/326) | Positive | IEDB ID: 1075117 1075077 [59] [In silico predicted, 22] | 6 |
| | 7 | YYVGYLQPR | 265–273 | HLA-A*33:01, HLA-A*31:01, HLA-A*68:01 | 3 | 1, 2 & 3 | Antigen (1.4692) | Non-allergen | 100.00% (175/175) | 0.31% (1/326) | Positive | IEDB ID: 1075077 [59] | 7 |
| | 8 | GYLQPRTFL | 268–276 | HLA-A*23:01, HLA-A*24:02 | 2 | 2 | Antigen (0.6082) | Non-allergen | 100.00% (175/175) | 0.31% (1/326) | Positive | [17, 76] | 8 |
| | 9 | AVDCALDPL | 288–296 | HLA-A*02:06 | 1 | 1 & 3 | Antigen (0.6604) | Non-allergen | 100.00% (175/175) | 0.31% (1/326) | Positive | - | 9 |
| | 10 | QIAPGQTGK | 409–417 | HLA-A*68:01, HLA-A*11:01, HLA-A*03:01 | 3 | 1, 2 & 3 | Antigen (1.8297) | Non-allergen | 100.00% (175/175) | 0.31% (1/326) | Positive | IEDB ID: 1075039 [59] | 10 |
| | 11 | TGKIADYNY | 415–423 | HLA-A*30:02 | 1 | 2 | Antigen (1.5305) | Non-allergen | 100.00% (175/175) | 0.31% (1/326) | Positive | IEDB ID: 1075039 [59] | 11 |
| | 12 | LDSKVGGNY | 441–449 | HLA-A*01:01 | 1 | 2 | Antigen (0.7814) | Non-allergen | 100.00% (175/175) | 0.00% (0/326) | Positive | IEDB ID: 1075002 [59] | 12 |
| | 13 | SKVGGNYNY | 443–451 | HLA-A*30:02, HLA-B*15:01 | 2 | 2 | Antigen (0.9111) | Non-allergen | Spike (S) | 0.00% (0/326) | Positive | IEDB ID: 1074979 [59] | 13 |

(Continued)

**Table 6.** (Continued)

| SARS-CoV-2 protein | Sl. No. | Peptide | Position | Alleles | Number of Alleles | Servers | Probable Antigenicity by Vexijen (Threshold = 0.4) | Probable Allergenicity by AllerTop | Conservancy analysis by IEDB tool | Conservancy analysis by IEDB tool — SARS-CoV-2 | SARS-CoV | IFN gamma epitope | Experimentally verified epitopes / ViPR reference |
|---|---|---|---|---|---|---|---|---|---|---|---|---|---|
| | 14 | YQPYRVVVL | 505–513 | HLA-A*02:06, HLA-B*08:01, HLA-A*02:03 | 3 | 1, 2 & 3 | Antigen (0.5964) | Non-allergen | 100.00% (175/175) | 84.97% (277/326) | Positive | [In silico predicted, 21, 22] [17] | 14 |
| | 15 | VVVLSFELL | 510–518 | HLA-A*02:06 | 1 | 1 & 3 | Antigen (1.0909) | Non-allergen | 100.00% (175/175) | 85.89% (280/326) | Positive | - | 15 |
| | 16 | GVLTESNKK | 550–558 | HLA-A*11:01, HLA-A*03:01 | 2 | 1, 2 & 3 | Antigen (0.8797) | Non-allergen | 100.00% (175/175) | 0.31% (1/326) | Positive | - | 16 |
| | 17 | CSFGGVSVI | 590–598 | HLA-A*68:02, HLA-A*32:01, HLA-A*02:06, HLA-A*02:03, HLA-B*58:01 | 5 | 1 & 3 | Antigen (1.2630) | Non-allergen | 100.00% (175/175) | 83.44% (272/326) | Positive | - | 17 |
| | 18 | SIAIPTNFT | 711–719 | HLA-A*68:02 | 1 | 1 | Antigen (0.5808) | Non-allergen | 100.00% (175/175) | 0.31% (1/326) | Positive | IEDB ID: 1074855 [59] | 18 |
| | 19 | GSFCTQLNR | 757–765 | HLA-A*11:01, HLA-A*31:01, HLA-A*68:01, HLA-A*03:01 | 4 | 1, 2 & 3 | Antigen (0.9306) | Non-allergen | 100.00% (175/175) | 76.07% (248/326) | Positive | [17] | 19 |
| | 20 | TLADAGFIK | 827–835 | HLA-A*11:01, HLA-A*68:01, HLA-A*03:01 | 3 | 1, 2 & 3 | Antigen (0.5781) | Non-allergen | 100.00% (175/175) | 0.31% (1/326) | Positive | IEDB ID: 1074948 [59] | 20 |
| | 21 | YTSALLAGT | 873–881 | HLA-A*68:02, HLA-A*02:06, HLA-A*02:03 | 3 | 1 & 3 | Antigen (0.5487) | Non-allergen | 100.00% (175/175) | 0.31% (1/326) | Positive | [17] | 21 |
| | 22 | WTFGAGAAL | 886–894 | HLA-A*68:02, HLA-A*02:06, HLA-B*35:01, HLA-B*15:01, HLA-A*02:03, HLA-A*26:01 | 6 | 1, 2 & 3 | Antigen (0.4918) | Non-allergen | 100.00% (175/175) | 85.28% (278/326) | Positive | IEDB ID: 1074998 [59] [In silico predicted, 20] | 22 |
| | 23 | FAMQMAYRF | 898–906 | HLA-B*35:01, HLA-B*53:01, HLA-B*58:01, HLA-A*23:01, HLA-B*15:01, HLA-A*24:02, HLA-A*32:01, HLA-B*57:01, HLA-A*02:06 | 9 | 1, 2 & 3 | Antigen (1.0278) | Non-allergen | 100.00% (175/175) | 86.20% (281/326) | Positive | [In silico predicted, 20, 22] | 23 |
| | 24 | SALGKLQDV | 943–951 | HLA-A*02:06 | 1 | 1 & 3 | Antigen (0.8291) | Non-allergen | 100.00% (175/175) | 0.31% (1/326) | Positive | - | 24 |
| | 25 | YIKWPWYIW | 1209–1217 | HLA-A*32:01, HLA-A*23:01, HLA-B*58:01, HLA-B*57:01 | 4 | 1 & 3 | Antigen (0.9673) | Non-allergen | 94.86% (166/175) | 0.92% (3/326) | Positive | [17] | 25 |
| | 26 | GLIAIVMVT | 1223–1231 | HLA-A*02:03, HLA-A*02:01 | 2 | 1, 2 & 3 | Antigen (1.0885) | Non-allergen | 94.86% (166/175) | 79.14% (258/326) | Positive | - | 26 |

(Continued)

**Table 6.** (Continued)

| SARS-CoV-2 protein | Sl. No. | Peptide | Position | Alleles | Number of Alleles | Servers | Probable Antigenicity by Vexilen (Threshold = 0.4) | Probable Allergenicity by AllerTop | Conservancy analysis by IEDB tool | Conservancy analysis by IEDB tool | | IFN gamma epitope | Experimentally verified epitopes / ViPR reference |
|---|---|---|---|---|---|---|---|---|---|---|---|---|---|
| | | | | | | | | | | SARS-CoV-2 | SARS-CoV | | |
| Nucleocapsid (N) | 1 | RSGARSKQR | 32–40 | HLA-A*31:01 | 1 | 1,2&3 | Antigen (0.9734) | Non-allergen | 99.46% (184/185) | 0.00% (0/301) | Positive | IEDB ID: 1075022 [59] | 1 |
| | 2 | FPRGQGVPI | 66–74 | HLA-B*07:02, HLA-B*35:01, HLA-B*51:01, HLA-B*08:01, HLA-B*53:01 | 5 | 1,2&3 | Antigen (1.2832) | Non-allergen | 100.00% (185/185) | 83.39% (251/301) | Positive | [61] | 2 |
| | 3 | SPDDQIGYY | 79–87 | HLA-B*35:01, HLA-B*53:01, HLA-A*01:01, HLA-A*30:02 | 4 | 1,2&3 | Antigen (0.864) | Non-allergen | 100.00% (185/185) | 0.00% (0/301) | Positive | IEDB ID: 1074917 [59] | 3 |
| | 4 | KDLSPRWYF | 102–110 | HLA-B*44:02, HLA-B*44:03 | 2 | 2 | Antigen (0.8611) | Non-allergen | 99.46% (184/185) | 0.00% (0/301) | Positive | [17], [In silico predicted, 23] | 4 |
| | 5 | KGFYAEGSR | 169–177 | HLA-A*31:01 | 1 | 1,2&3 | Antigen (0.4353) | Non-allergen | 100.00% (185/185) | 85.71% (258/301) | Positive | - | 5 |
| | 6 | RGGSQASSR | 177–185 | HLA-A*31:01 | 1 | 1,2&3 | Antigen (0.7468) | Non-allergen | 100.00% (185/185) | 79.73% (240/301) | Positive | - | 6 |
| | 7 | QASSRSSSR | 181–189 | HLA-A*68:01, HLA-A*31:01 | 2 | 1,2&3 | Antigen (0.5495) | Non-allergen | 100.00% (185/185) | 81.40% (245/301) | Positive | - | 7 |
| | 8 | SSRNSTPGS | 193–201 | HLA-A*30:01 | 1 | 1,2&3 | Antigen (0.6287) | Non-allergen | 100.00% (185/185) | 0.66% (2/301) | Positive | - | 8 |
| | 9 | SSRGTSPAR | 201–209 | HLA-A*31:01, HLA-A*30:01, HLA-A*68:01 | 3 | 1,2&3 | Antigen (0.4548) | Non-allergen | 100.00% (185/185) | 1.33% (4/301) | Positive | IEDB ID: 1075062 [59] | 9 |
| | 10 | KSAAEASKK | 249–257 | HLA-A*11:01, HLA-A*03:01, HLA-A*30:01 | 3 | 1,2&3 | Antigen (0.5547) | Non-allergen | 100.00% (185/185) | 83.39% (251/301) | Positive | - | 10 |
| | 11 | KAYNVTQAF | 266–274 | HLA-A*32:01, HLA-B*58:01, HLA-B*15:01, HLA-B*57:01, HLA-B*35:01, HLA-A*30:02, HLA-B*53:01, HLA-A*23:01, HLA-A*30:01, HLA-B*07:02, HLA-A*24:02 | 11 | 1,2&3 | Antigen (1.7645) | Non-allergen | 100.00% (185/185) | 0.66% (2/301) | Positive | IEDB ID: 1074947 [59] | 11 |
| | 12 | QFAPSASAF | 306–314 | HLA-A*24:02, HLA-A*23:01, HLA-B*15:01, HLA-B*35:01 | 4 | 1&2 | Antigen (0.55) | Non-allergen | 100.00% (185/185) | 85.71% (258/301) | Positive | [In silico predicted, 23] | 12 |
| | 13 | ASAFFGMSR | 311–319 | HLA-A*11:01, HLA-A*68:01, HLA-A*31:01, HLA-A*03:01, HLA-A*30:01, HLA-A*33:01 | 6 | 1,2&3 | Antigen (0.4529) | Non-allergen | 100.00% (185/185) | 85.71% (258/301) | Positive | IEDB ID: 56979 [59] | 13 |

(Continued)

**Table 6.** (Continued)

| SARS-CoV-2 protein | Sl. No. | Peptide | Position | Alleles | Number of Alleles | Servers | Probable Antigenicity by Vexijen (Threshold = 0.4) | Probable Allergenicity by AllerTop | Conservancy analysis by IEDB tool | Conservancy analysis by IEDB tool SARS-CoV-2 | SARS-CoV | IFN gamma epitope | Experimentally verified epitopes / ViPR reference |
|---|---|---|---|---|---|---|---|---|---|---|---|---|---|
| | 14 | DPNFKDQVI | 343–351 | HLA-B*51:01, HLA-B*08:01, HLA-B*53:01 | 3 | 2 | Antigen (0.5439) | Non-allergen | 100.00% (185/185) | 0.33% (1/301) | Positive | [17] | 14 |
| | 15 | LNKHIDAYK | 353–361 | HLA-A*68:01, HLA-A*31:01, HLA-A*30:01 | 3 | 3 | Antigen (1.7367) | Non-allergen | 98.38% (182/185) | 85.71% (258/301) | Positive | - | 15 |
| | 16 | KTFPPTEPK | 361–369 | HLA-A*11:01, HLA-A*30:01, HLA-A*03:01, HLA-A*31:01, HLA-A*68:01 | 5 | 1,2&3 | Antigen (1.1677) | Non-allergen | 100.00% (185/185) | 84.05% (253/301) | Positive | - | 16 |
| Membrane (M) | 1 | GTITVEELK | 6–14 | HLA-A*11:01, HLA-A*68:01 | 2 | 2&3 | Antigen (1.0976) | Non-allergen | 98.82% (168/170) | 50.00% (134/268) | Positive | IEDB ID: 107499 [59] | 1 |
| | 2 | WICLLQFAY | 31–39 | HLA-A*02:06, HLA-A*30:01 | 2 | 3 | Antigen (1.4562) | Non-allergen | 98.82% (168/170) | 0.75% (2/268) | Positive | IEDB ID: 10749920 [59] | 2 |
| | 3 | AYANRNRFL | 38–46 | HLA-A*02:06, HLA-A*02:01, HLA-A*68:02, HLA-A*02:03, HLA-A*32:01 | 5 | 1,2&3 | Antigen (0.5136) | Non-allergen | 98.24% (167/170) | 0.75% (2/268) | Positive | IEDB ID: 1074985 [59] | 3 |
| | 4 | LWLLWPVTL | 54–62 | HLA-B*40:01, HLA-B*44:03, HLA-B*44:02 | 3 | 2&3 | Antigen (0.6409) | Non-allergen | 98.82% (168/170) | 82.46% (221/268) | Positive | [76] | 4 |
| | 5 | CFVLAAVYR | 64–72 | HLA-A*33:01, HLA-A*31:01 | 2 | 1,2&3 | Antigen (0.9181) | Non-allergen | 98.82% (168/170) | 78.73% (211/268) | Positive | - | 5 |
| | 6 | WITGGIAIA | 75–83 | HLA-A*02:01, HLA-A*02:06 | 2 | 1&3 | Antigen (1.4835) | Non-allergen | 98.82% (168/170) | 0.75% (2/268) | Positive | IEDB ID: 1075124 [59] | 6 |
| | 7 | MACLVGLMW | 84–92 | HLA-A*33:01, HLA-A*31:01, HLA-A*68:01 | 3 | 1,2&3 | Antigen (1.1961) | Non-allergen | 98.82% (168/170) | 0.75% (2/268) | Positive | IEDB ID: 1075124 [59] | 7 |
| | 8 | FIASFRLFA | 96–104 | HLA-A*32:01, HLA-B*58:01, HLA-A*23:01, HLA-B*57:01 | 4 | 1&3 | Antigen (0.4968) | Non-allergen | 98.24% (167/170) | 3.36% (9/268) | Positive | IEDB ID: 1075069 [59] | 8 |
| | 9 | RSMWSFNPE | 107–115 | HLA-A*02:06, HLA-A*68:02, HLA-B*58:01, HLA-A*02:03, HLA-B*51:01 | 5 | 3 | Antigen (1.1704) | Non-allergen | 98.82% (168/170) | 82.46% (221/268) | Positive | IEDB ID: 1075081 [59] | 9 |
| | 10 | TSRTLSYYK | 172–180 | HLA-A*68:01, HLA-A*31:01, HLA-A*11:01, HLA-A*33:01 | 4 | 1,2&3 | Antigen (1.1027) | Non-allergen | 98.82% (168/170) | 80.97% (217/268) | Positive | [17] | 10 |

(Continued)

**Table 6.** (Continued)

| SARS-CoV-2 protein | Sl. No. | Peptide | Position | Alleles | Number of Alleles | Servers | Probable Antigenicity by VexiJen (Threshold = 0.4) | Probable Allergenicity by AllerTop | Conservancy analysis by IEDB tool | Conservancy analysis by IEDB tool SARS-CoV-2 | SARS-CoV | IFN gamma epitope | Experimentally verified epitopes / ViPR reference |
|---|---|---|---|---|---|---|---|---|---|---|---|---|---|
| Envelope (E) | 1 | VLLFLAFVV | 17–25 | HLA-A*02:01, HLA-A*02:06, HLA-A*02:03, HLA-A*32:01 | 4 | 1 & 3 | Antigen (0.5677) | Non-allergen | 100.00% (177/177) | 83.27% (204/245) | Positive | IEDB ID: 62215 [59] | 1 |
| | 2 | FLLVTLAIL | 26–34 | HLA-A*02:01, HLA-A*02:03, HLA-A*02:06 | 3 | 1 & 3 | Antigen (0.9645) | Non-allergen | 100.00% (177/177) | 86.12% (211/245) | Positive | - | 2 |
| | 3 | RLCAYCCNI | 38–46 | HLA-A*02:03, HLA-A*02:01, HLA-A*02:06 | 3 | 3 | Antigen (1.1243) | Non-allergen | 100.00% (177/177) | 84.90% (208/245) | Positive | IEDB ID: 154490 [59] | 3 |

List of top-ranked CD8+ T-cell epitopes predicted using at least two servers, with the corresponding prediction servers numbered as (1) for IEDB Proteasomal cleavage/TAP transport/MHC class I combined predictor, (2) for MHC-NP, and (3) for TepiTool. Epitopes of 9-mer length with IC50 value < 500 nM were considered good binders towards specific alleles.

Table 7. List of predicted T-cell epitopes with high-affinity binding to MHC I molecules common to SARS-CoV-2 and SARS-CoV.

| CoV protein | Sl. No. | Peptide | SARS-CoV-2 | | | | | SARS-CoV | | | | | Probable Antigenicity by Vexijen (Threshold = 0.4) | Probable Allergenicity by AllerTop | IFN gamma epitope |
|---|---|---|---|---|---|---|---|---|---|---|---|---|---|---|---|
| | | | Position | Alleles | Number of Alleles | Servers | Conservancy | Position | Alleles | Number of Alleles | Servers | Conservancy | | | |
| Spike (S) | 1 | PYRVVVLSF | 507–515 | HLA-A*23:01, HLA-A*24:02 | 2 | 1, 2 & 3 | 100.00% (175/175) | 493–501 | HLA-A*23:01, HLA-A*24:02 | 2 | 1, 2 & 3 | 85.89% (280/326) | Antigen (1.0281) | Non-Allergen | Positive |
| | 2 | VVVLSFELL | 510–518 | HLA-A*02:06 | 1 | 1 & 3 | 100.00% (175/175) | 496–504 | HLA-A*02:06 | 1 | 1 & 3 | 85.89% (280/326) | Antigen (1.0909) | Non-Allergen | Positive |
| | 3 | WTFGAGAAL | 886–894 | HLA-A*68:02, HLA-A*02:06, HLA-B*35:01, HLA-B*15:01, HLA-A*26:01, HLA-A*02:03 | 6 | 1, 2 & 3 | 100.00% (175/175) | 868–876 | HLA-A*68:02, HLA-A*02:06, HLA-B*35:01, HLA-B*15:01, HLA-A*26:01, HLA-A*02:03 | 6 | 1, 2 & 3 | 85.28% (278/326) | Antigen (0.4918) | Non-Allergen | Positive |
| | 4 | FAMQMAYRF | 898–909 | HLA-B*35:01, HLA-B*53:01, HLA-B*58:01, HLA-A*23:01, HLA-A*24:02, HLA-B*15:01, HLA-A*32:01, HLA-A*02:06, HLA-B*51:01, HLA-B*57:01, HLA-A*02:06 | 11 | 1, 2 & 3 | 100.00% (175/175) | 880–888 | HLA-B*35:01, HLA-B*53:01, HLA-B*58:01, HLA-A*23:01, HLA-A*24:02, HLA-B*15:01, HLA-A*32:01, HLA-A*02:06, HLA-B*57:01, HLA-B*08:01, HLA-B*51:01 | 11 | 1, 2 & 3 | 86.20% (281/326) | Antigen (1.0278) | Non-Allergen | Positive |
| | 5 | GLIAIVMVT | 1223–1231 | HLA-A*02:03, HLA-A*02:01 | 2 | 1, 2 & 3 | 94.86% (166/175) | 1205–1213 | HLA-A*02:03, HLA-A*02:01 | 2 | 2 & 3 | 79.14% (258/326) | Antigen (1.0885) | Non-Allergen | Positive |
| Nucleocapsid (N) | 1 | LSPRWYFYY | 104–112 | HLA-A*30:02, HLA-A*01:01, HLA-B*58:01, HLA-A*11:01 | 4 | 1,2 & 3 | 100.00% (185/185) | 105–113 | HLA-A*30:02, HLA-A*01:01, HLA-B*58:01, HLA-A*11:01 | 4 | 1, 2 & 3 | 85.71% (258/301) | Antigen (1.2832) | Non-Allergen | Positive |
| | 2 | RGGSQASSR | 177–185 | HLA-A*31:01 | 1 | 1 & 2 | 99.46% (184/185) | 178–186 | HLA-A*31:01 | 1 | 1, 2 & 3 | 79.73% (240/301) | Antigen (0.8611) | Non-Allergen | Positive |
| | 3 | QFAPSASAF | 306–314 | HLA-A*24:02, HLA-A*23:01, HLA-B*15:01, HLA-B*35:01 | 4 | 1, 2 & 3 | 100.00% (185/185) | 307–315 | HLA-A*24:02, HLA-A*23:01, HLA-B*15:01, HLA-B*35:01 | 4 | 1 & 2 | 85.71% (258/301) | Antigen (0.5495) | Non-Allergen | Positive |
| | 4 | GMSRIGMEV | 316–324 | HLA-A*02:03, HLA-A*02:01, HLA-A*02:06 | 3 | 1, 2 & 3 | 100.00% (185/185) | 317–325 | HLA-A*02:03, HLA-A*02:01, HLA-A*02:06 | 3 | 1, 2 & 3 | 84.72% (255/301) | Antigen (0.6287) | Non-Allergen | Positive |
| | 5 | EVTPSGTWL | 323–331 | HLA-A*68:02 | 1 | 1, 2 & 3 | 100.00% (185/185) | 324–332 | HLA-A*68:02 | 1 | 1, 2 & 3 | 83.72% (252/301) | Antigen (0.4548) | Non-Allergen | Positive |
| Membrane (M) | 1 | GTITVEELK | 6–14 | HLA-A*11:01, HLA-A*68:01 | 2 | 2 & 3 | 98.82% (168/170) | 5–13 | HLA-A*11:01, HLA-A*68:01 | 2 | 2&3 | 50.00% (134/268) | Antigen (1.0976) | Non-Allergen | Positive |
| | 2 | LWLLWPVTL | 54–62 | HLA-A*23:01, HLA-A*24:02 | 2 | 1,2 & 3 | 98.24% (167/170) | 53–61 | HLA-A*23:01, HLA-A*24:02 | 2 | 1,2 & 3 | 81.34% (218/268) | Antigen (1.1072) | Non-Allergen | Positive |
| | 3 | RYRIGNYKL | 198–206 | HLA-A*30:01, HLA-A*23:01, HLA-A*24:02 | 3 | 1, 2 & 3 | 98.82% (168/170) | 197–205 | HLA-A*30:01, HLA-A*23:01, HLA-A*24:02 | 3 | 1, 2 & 3 | 82.46% (221/268) | Antigen (0.9181) | Non-Allergen | Positive |

(*Continued*)

**Table 7.** (Continued)

| CoV protein | Sl. No. | Peptide | SARS-CoV-2 | | | | | SARS-CoV | | | | | Probable Antigenicity by Vexijen (Threshold = 0.4) | Probable Allergenicity by AllerTop | IFN gamma epitope |
|---|---|---|---|---|---|---|---|---|---|---|---|---|---|---|---|
| | | | Position | Alleles | Number of Alleles | Servers | Conservancy | Position | Alleles | Number of Alleles | Servers | Conservancy | | | |
| Envelope (E) | 1 | SVLLFLAFV | 16–24 | HLA-A*02:06, HLA-A*02:01, HLA-A*68:02, HLA-A*02:03 | 4 | 1, 2 & 3 | 100.00% (177/177) | 16–24 | HLA-A*02:06, HLA-A*02:01, HLA-A*68:02, HLA-A*02:03 | 4 | 1 & 3 | 83.27% (204/245) | Antigen (0.4765) | Non-Allergen | Positive |
| | 2 | FLAFVVFLL | 20–28 | HLA-A*02:01, HLA-A*02:03, HLA-A*02:06, HLA-A*68:02 | 4 | 1, 2 & 3 | 100.00% (177/177) | 20–28 | HLA-A*02:01, HLA-A*02:03, HLA-A*02:06, HLA-A*68:02 | 4 | 1, 2 & 3 | 83.67% (205/245) | Antigen (0.5308) | Non-Allergen | Positive |
| | 3 | FLLVTLAIL | 26–34 | HLA-A*02:01, HLA-A*02:03, HLA-A*02:06 | 3 | 1 & 3 | 100.00% (177/177) | 26–34 | HLA-A*02:01, HLA-A*02:03, HLA-A*02:06 | 3 | 1 & 3 | 86.12% (211/245) | Antigen (0.9645) | Non-Allergen | Positive |

List of top-ranked CD8+ T-cell epitopes predicted using at least two servers, with the corresponding prediction servers numbered as (1) for IEDB Proteasomal cleavage/TAP transport/MHC class I combined predictor, (2) for MHC-NP, and (3) for TepiTool. Epitopes of 15-mer length with IC50 value <1000 nM were considered good binders towards specific alleles.

**Table 8. List of predicted SARS-CoV-2-derived T-cell epitopes with high affinity binding to MHC II molecules.**

| SARS-CoV-2 protein | Sl. No. | Peptide | Core | Position | Alleles | Number of Alleles | Servers | Probable Antigenicity by VexiJen (Threshold = 0.4) | Probable Allergenicity by AllerTop | Conservancy (SARS-CoV-2) | Conservancy (SARS-CoV) | IFN gamma epitope | Experimentally verified epitopes / ViPR reference |
|---|---|---|---|---|---|---|---|---|---|---|---|---|---|
| Spike (S) | 1 | AGAAAYYVGYLQPRT | - | 260–274 | HLA-DQA1*01:01/DQB1*05:01, HLA-DPA1*01:03/DPB1*02:01, HLA-DPA1*01:03/DPB1*04:01, HLA-DPA1*02:01/DPB1*01:01, HLA-DRB1*15:01, HLA-DPA1*03:01/DPB1*04:02, HLA-DQA1*05:01/DQB1*03:01, HLA-DRB5*01:01, HLA-DRB1*11:01, HLA-DRB1*04:01, HLA-DRB1*01:01, HLA-DRB1*07:01 | 14 | 2 | Antigen (0.9134) | Non-allergen | 0.31% (1/326) | 100.00% (175/175) | Positive | - |
| | 2 | KGIYQTSNFRVQPTE | IYQTSNFRV YQTSNFRVQ | 310–324 | HLA-DRB1*07:01, HLA-DRB1*13:02, HLA-DRB1*15:01, HLA-DPA1*02:01/DPB1*01:01, HLA-DPA1*01:03/DPB1*02:01, HLA-DPA1*01:03/DPB1*04:01, HLA-DPA1*03:01/DPB1*04:02, HLA-DPA1*02:01/DPB1*05:01 | 8 | 1 | Antigen (0.8838) | Non-allergen | 0.00% (0/326) | 100.00% (175/175) | Positive | [In silico predicted from aa 309–323, 27] |
| | 3 | FSTFKCYGVSPTKLN | - | 374–388 | HLA-DRB1*09:01, HLA-DRB1*07:01, HLA-DRB1*01:01, HLA-DRB1*04:01, HLA-DRB1*04:05, HLA-DRB1*15:01, HLA-DQA1*05:01/DQB1*03:01, HLA-DRB1*11:01 | 8 | 2 | Antigen (1.0042) | Non-allergen | 0.61% (2/326) | 100.00% (175/175) | Positive | - |
| | 4 | VLSFELLHAPATVCG | FELLHAPAT | 512–526 | HLA-DRB1*01:01, HLA-DRB1*09:01, HLA-DRB1*04:01, HLA-DRB1*04:05, HLA-DRB1*07:01, HLA-DRB1*08:02, HLA-DPA1*03:01/DPB1*04:02, HLA-DRB3*02:02, HLA-DPA1*01:03/DPB1*04:01, HLA-DPA1*02:01/DPB1*01:01, HLA-DRB1*15:01, HLA-DPA1*01:03/DPB1*02:01, HLA-DQA1*05:01/DQB1*03:01, HLA-DQA1*01:02/DQB1*06:02, HLA-DRB1*11:01, HLA-DRB5*01:01, HLA-DRB1*13:02, HLA-DRB4*01:01 | 18 | 1 & 2 | Antigen (0.4874) | Non-allergen | 0.00% (0/326) | 100.00% (175/175) | Positive | [In silico predicted, 22] |
| | 5 | TGVLTESNKKFLPFQ | - | 549–563 | HLA-DRB1*03:01, HLA-DRB1*13:02, HLA-DRB5*01:01, HLA-DRB1*01:01 | 4 | 2 | Antigen (0.8330) | Non-allergen | 0.31% (1/326) | 100.00% (175/175) | Positive | - |
| | 6 | PVAIHADQLTPTWRV | - | 621–635 | HLA-DRB1*03:01, HLA-DRB3*01:01, HLA-DRB1*13:02, HLA-DRB1*04:01, HLA-DRB1*07:01, HLA-DRB4*01:01, HLA-DRB1*01:01 | 7 | 2 | Antigen (0.5235) | Non-allergen | 0.31% (1/326) | 100.00% (175/175) | Positive | - |
| | 7 | VYSTGSNVFQTRAGC | STGSNVFQT | 635–648 | HLA-DQA1*01:02/DQB1*06:02 | 1 | 1 | Antigen (0.5659) | Non-allergen | 0.31% (1/326) | 100.00% (175/175) | Positive | - |
| | 8 | ICASYQTQTNSPRRA | YQTQTNSPR | 670–684 | HLA-DRB1*04:01, HLA-DRB1*04:05, HLA-DRB5*01:01, HLA-DRB1*01:01 | 4 | 1 & 2 | Antigen (0.2829) | Non-allergen | 0.00% (0/326) | 94.86% (166/175) | Positive | - |
| | 9 | VASQSIIAYTMSLGA | - | 687–701 | HLA-DQA1*01:02/DQB1*06:02, HLA-DRB1*07:01, HLA-DRB1*15:01, HLA-DRB1*09:01, HLA-DPA1*01:03/DPB1*02:01, HLA-DRB1*01:01, HLA-DRB1*04:05, HLA-DRB1*12:01, HLA-DQA1*05:01/DQB1*03:01, HLA-DRB1*08:02, HLA-DPA1*01:03/DPB1*04:01, HLA-DPA1*03:01/DPB1*04:02, HLA-DRB1*04:01, HLA-DRB4*01:01, HLA-DPA1*02:01/DPB1*01:01, HLA-DRB1*13:02, HLA-DRB1*11:01, HLA-DRB5*01:01 | 18 | 2 | Antigen (0.5204) | Non-allergen | 0.31% (1/326) | 100.00% (175/175) | Positive | - |

(Continued)

**Table 8.** (Continued)

| SARS-CoV-2 protein | Sl. No. | Peptide | Core | Position | Alleles | Number of Alleles | Servers | Probable Antigenicity by VexiJen (Threshold = 0.4) | Probable Allergenicity by AllerTop | Conservancy (SARS-CoV-2) | Conservancy (SARS-CoV) | IFN gamma epitope | Experimentally verified epitopes / ViPR reference |
|---|---|---|---|---|---|---|---|---|---|---|---|---|---|
| Spike (S) | 10 | LLQYGSFCTQLNRAL | - | 753–767 | HLA-DRB1*04:05, HLA-DRB1*04:01, HLA-DRB1*09:01, HLA-DPA1*01:03/DPB1*02:01, HLA-DPA1*01:03/DPB1*04:01, HLA-DRB1*01:01, HLA-DRB1*15:01, HLA-DPA1*02:01/DPB1*01:01, HLA-DRB1*07:01, HLA-DPA1*03:01/DPB1*04:02, HLA-DRB5*01:01, HLA-DRB1*11:01, HLA-DRB1*08:02/DQB1*06:02, HLA-DRB1*08:02, HLA-DQA1*05:01/DQB1*03:01, HLA-DRB4*01:01 | 16 | 2 | Antigen (0.722) | Non-allergen | 76.07% (248/326) | 100.00% (175/175) | Positive | [In silico predicted, 22] |
| | 11 | ALTGIAVEQDKNTQE | - | 766–780 | HLA-DQA1*05:01/DQB1*02:01, HLA-DQA1*01:02/DQB1*06:02 | 2 | 2 | Antigen (0.6953) | Non-allergen | 0.31% (1/326) | 100.00% (175/175) | Positive | - |
| | 12 | VKQLSSNFGAISSVL | - | 963–977 | HLA-DRB1*13:02, HLA-DQA1*01:02/DQB1*06:02, HLA-DQA1*05:01/DQB1*03:01, HLA-DRB1*09:01, HLA-DRB1*04:01, HLA-DRB1*01:01, HLA-DRB1*07:01, HLA-DRB1*15:01, HLA-DRB1*04:05, HLA-DPA1*01:03/DPB1*02:01, HLA-DRB1*08:02, HLA-DPA1*02:01/DPB1*01:01, HLA-DRB1*11:01, HLA-DRB5*01:01, HLA-DRB4*01:01 | 15 | 2 | Antigen (0.4784) | Non-allergen | 85.89% (280/326) | 100.00% (175/175) | Positive | [In silico predicted, 22] |
| | 13 | RAAEIRASANLAATK | IRASANLAA | 1014–1028 | HLA-DRB1*04:01, HLA-DRB1*08:02, HLA-DRB1*13:02, HLA-DRB1*15:01, HLA-DRB3*02:02, HLA-DRB4*01:01, HLA-DPA1*02:01/DPB1*14:01 | 7 | 1 | Antigen (0.5709) | Non-allergen | 85.58% (279/326) | 100.00% (175/175) | Positive | IEDB ID: 100428 [59] [In silico predicted, 20, 21] |
| | 14 | VVFLHVTYVPAQEKN | VTYVPAQEK FLHVTYVPA | 1060–1074 | HLA-DRB5*01:01, HLA-DPA1*02:01/DPB1*01:01, HLA-DPA1*01:03/DPB1*02:01, HLA-DPA1*01:03/DPB1*04:01, HLA-DPA1*03:01/DPB1*04:02, HLA-DPA1*02:01/DPB1*05:01 | 6 | 1 | Antigen (1.172) | Non-allergen | 0.31% (1/326) | 100.00% (175/175) | Positive | [In silico predicted, 27] |
| Nucleocapsid (N) | 1 | IGYYRRATRRIRGGD | YRRATRRIR | 84–98 | HLA-DRB1*08:02, HLA-DRB1*11:01, HLA-DRB1*13:02 | 3 | 1 | Antigen (0.6649) | Non-allergen | 100.00% (185/185) | 0.66% (2/301) | Positive | [77] |
| | 2 | MKDLSPRWYFYYLGT | - | 101–115 | HLA-DPA1*01:03/DPB1*04:01, HLA-DPA1*01:03/DPB1*02:01, HLA-DPA1*02:01/DPB1*01:01, HLA-DRB1*15:01 | 4 | 2 | Antigen (1.0027) | Non-allergen | 100.00% (185/185) | 0.00% (0/301) | Positive | - |
| | 3 | YKHWPQIAQFAPSAS | - | 298–312 | HLA-DQA1*05:01/DQB1*03:01, HLA-DQA1*01:02/DQB1*06:02, HLA-DRB1*09:01, HLA-DRB1*01:01, HLA-DRB1*04:01, HLA-DRB1*04:05 | 6 | 1 & 2 | Antigen (0.434) | Non-allergen | 100.00% (185/185) | 82.72% (249/301) | Positive | [77] |
| | 4 | QIAQFAPSASAFFGM | FAPSASAFF | 303–317 | HLA-DRB1*09:01, HLA-DRB1*07:01, HLA-DQA1*05:01/DQB1*03:01, HLA-DPA1*01:01, HLA-DPA1*02:01/DPB1*14:01, HLA-DPA1*01:02/DQA1*01:02/DPB1*04:05, HLA-DRB1*06:02, HLA-DRB3*01:01, HLA-DRB1*08:02, HLA-DRB1*15:01, HLA-DRB1*13:02, HLA-DPA1*01:03/DPB1*02:01, HLA-DRB5*01:01, HLA-DPA1*01:03/DPB1*01:01, HLA-DPA1*04:01, HLA-DRB4*01:01, HLA-DRB1*11:01 | 19 | 2 | Antigen (0.4032) | Non-allergen | 100.00% (185/185) | 85.71% (258/301) | Positive | - |
| | 5 | PSASAFFGMSRIGME | FFGMSRIGM | 309–323 | HLA-DRB1*11:01 | 1 | 1 | Antigen (0.6408) | Non-allergen | 100.00% (185/185) | 85.05% (256/301) | Positive | IEDB ID: 3958 [59] |
| | 6 | NFKDQVILLNKHIDA | VILLNKHID | 345–359 | HLA-DRB1*08:02, DRB1*11:01, HLA-DRB1*12:01 | 3 | 1 | Antigen (0.7693) | Non-allergen | 100.00% (185/185) | 0.33% (1/301) | Positive | - |

(Continued)

**Table 8.** (Continued)

| SARS-CoV-2 protein | Sl. No. | Peptide | Core | Position | Alleles | Number of Alleles | Servers | Probable Antigenicity by VexiJen (Threshold = 0.4) | Probable Allergenicity by AllerTop | Conservancy (SARS-CoV-2) | Conservancy (SARS-CoV) | IFN gamma epitope | Experimentally verified epitopes / ViPR reference |
|---|---|---|---|---|---|---|---|---|---|---|---|---|---|
| Membrane (M) | 1 | LWLLWPVTLACFVLA | - | 54–68 | HLA-DQA1*01:01/DQB1*05:01, HLA-DPA1*01:03/DPB1*02:01, HLA-DPA1*01:03/DPB1*04:01, HLA-DPA1*03:01/DPB1*04:02, HLA-DPA1*02:01/DPB1*01:01, HLA-DRB1*04:05, HLA-DRB1*15:01, HLA-DRB1*07:01, HLA-DRB1*12:01, HLA-DRB1*01:01, HLA-DRB1*09:01, HLA-DQA1*01:02/DQB1*06:02, HLA-DRB1*04:01, HLA-DRB4*01:01 | 14 | 2 | Antigen (0.937) | Non-allergen | 82.46% (221/268) | 98.24% (167/170) | Positive | - |
| | 2 | PVTLACFVLAAVYRI | - | 59–73 | HLA-DPA1*01:03/DPB1*02:01, HLA-DPA1*01:03/DPB1*04:01, HLA-DPA1*03:01/DPB1*04:02, HLA-DQA1*01:02/DQB1*06:02, HLA-DQA1*01:01/DQB1*05:01, HLA-DRB1*09:01, HLA-DRB1*01:01, HLA-DRB1*15:01, HLA-DPA1*02:01/DPB1*01:01, HLA-DRB1*12:01, HLA-DRB3*01:01, HLA-DRB5*01:01, HLA-DRB1*04:05, HLA-DRB1*13:02, HLA-DRB1*11:01, HLA-DQA1*05:01/DQB1*03:01, HLA-DRB1*08:02, HLA-DRB1*04:01, HLA-DRB4*01:01, HLA-DRB1*07:01 | 20 | 2 | Antigen (0.8548) | Non-allergen | 78.73% (211/268) | 98.24% (167/170) | Positive | - |
| | 3 | LVIGAVILRGHLRIA | - | 138–152 | HLA-DRB4*01:01, HLA-DRB1*12:01, HLA-DRB1*15:01, HLA-DRB5*01:01, HLA-DRB1*11:01, HLA-DRB1*08:02, HLA-DQA1*01:02/DQB1*06:02, HLA-DPA1*02:01/DPB1*05:01, HLA-DRB1*03:01, HLA-DRB1*01:01, HLA-DRB1*13:02, HLA-DQA1*05:01/DQB1*03:01, HLA-DRB1*07:01, HLA-DRB1*09:01, HLA-DPA1*01:03/DPB1*02:01, HLA-DPA1*02:01/DPB1*01:01, HLA-DPA1*04:05, HLA-DRB1*04:01 | 24 | 2 | Antigen (0.8769) | Non-allergen | 0.75% (2/268) | 98.82% (168/170) | Positive | - |
| Envelope (E) | 1 | AFVVFLLVTLAILTA | - | 22–36 | HLA-DPA1*01:03/DPB1*02:01, HLA-DRB1*01:01 | 2 | 2 | Antigen (0.6229) | Non-allergen | 100.00% (177/177) | 0.00% (0/326) | Positive | - |

List of top-ranked CD4+ T-cell epitopes predicted using at least two servers, with the corresponding prediction servers numbered as (1) for NetMHCpan4 and (2) for TepiTool.

**Table 9. T-cell epitopes predicted with high affinity to MHC II molecules common to SARS-CoV-2 and SARS-CoV.**

| SARS-CoV-2 protein | Peptide | Sl. No. | SARS-CoV-2 Core | SARS-CoV-2 Position | SARS-CoV-2 Alleles | SARS-CoV-2 Number of Alleles | SARS-CoV-2 Conservancy | SARS-CoV-2 Servers | SARS-CoV Core | SARS-CoV Position | SARS-CoV Alleles | SARS-CoV Number of Alleles | SARS-CoV Conservancy | SARS-CoV Servers | Probable Antigenicity by VexiJen (Threshold = 0.4) | Probable Allergenicity by AllerTop | IFN gamma epitope | Experimentally verified epitopes / VIPR reference |
|---|---|---|---|---|---|---|---|---|---|---|---|---|---|---|---|---|---|---|
| Spike (S) | LLQYGSFCTQLNRAL | 1 | - | 753–767 | HLA-DRB1*04:05, HLA-DRB1*04:01, HLA-DRB1*09:01, HLA-DPA1*01:03/DPB1*02:01, HLA-DPA1*01:03/DPB1*04:01, HLA-DRB1*01:01, HLA-DRB1*15:01, HLA-DPA1*02:01/DPB1*01:01, HLA-DRB1*07:01, HLA-DPA1*03:01/DPB1*04:02, HLA-DRB5*01:01, HLA-DRB1*11:01, HLA-DQA1*01:02/DQB1*06:02, HLA-DRB1*08:02, HLA-DQA1*05:01/DQB1*03:01, HLA-DRB4*01:01 | 16 | 100.00% (175/175) | 2 | - | 735–749 | HLA-DRB1*04:05, HLA-DRB1*04:01, HLA-DRB1*09:01, HLA-DPA1*01:03/DPB1*02:01, HLA-DPA1*01:03/DPB1*04:01, HLA-DRB1*01:01, HLA-DRB1*15:01, HLA-DPA1*02:01/DPB1*01:01, HLA-DRB1*07:01, HLA-DPA1*03:01/DPB1*04:02, HLA-DRB5*01:01, HLA-DRB1*11:01, HLA-DQA1*01:02/DQB1*06:02, HLA-DRB1*08:02, HLA-DQA1*05:01/DQB1*03:01, HLA-DRB4*01:01 | 16 | 76.07% (248/326) | 2 | Antigen (0.722) | Non-allergen | Positive | - |
| | RAAEIRASANLAATK | 2 | IRASANLAA | 1014–1028 | HLA-DRB1*04:01, HLA-DRB1*08:02, HLA-DRB1*13:02, HLA-DRB1*15:01, HLA-DRB4*01:01, HLA-DPA1*02:01/DPB1*14:01 | 6 | 100.00% (175/175) | 1 | IRASANLAA | 996–1010 | HLA-DRB1*04:01, HLA-DRB1*13:02, HLA-DRB1*08:02, HLA-DRB1*15:01, HLA-DRB3*02:02, HLA-DRB4*01:01, HLA-DPA10201/DPB11401 | 7 | 85.58% (279/326) | 1 | Antigen (0.5709) | Non-allergen | Positive | IEDB-ID: 100428 [59] |
| | VNFNFNGLTGTGVLT | 3 | FNGLTGTGV | 539–553 | HLA-DRB1*09:01, HLA-DRB1*01:01, HLA-DRB1*07:01, HLA-DRB3*02:02, HLA-DQA1*05:01/DQB1*03:01, HLA-DRB1*04:01, HLA-DRB1*04:05, HLA-DRB1*13:02, HLA-DRB5*01:01, HLA-DRB1*15:01, HLA-DRB1*11:01 | 11 | 100.00% (175/175) | 1 & 2 | FNGLTGTGV | 525–539 | HLA-DRB1*09:01, HLA-DRB1*01:01, HLA-DRB1*07:01, HLA-DRB3*02:02, HLA-DQA1*05:01/DQB1*03:01, HLA-DRB1*04:01, HLA-DRB1*04:05, HLA-DRB1*13:02, HLA-DRB5*01:01, HLA-DRB1*15:01, HLA-DRB1*11:01 | 11 | 86.20% (281/326) | 1 & 2 | Antigen (1.2439) | Allergen | Negative | [59] |
| | AMQMAYRFNGIGVTQ | 4 | YRFNGIGVT | 899–913 | HLA-DRB3*02:02 | 1 | 100.00% (175/175) | 1 | YRFNGIGVT | 881–895 | HLA-DRB3*02:02 | 1 | 86.20% (281/326) | 1 | Antigen (1.2934) | Non-allergen | Negative | [43] |

**Table 9.** (Continued)

| SARS-CoV-2 protein | Sl. No. | Peptide | SARS-CoV-2 | | | | | | SARS-CoV | | | | | | Probable Antigenicity by Vexijen (Threshold = 0.4) | Probable Allergenicity by AllerTop | IFN gamma epitope | Experimentally verified epitopes / VIPR reference |
|---|---|---|---|---|---|---|---|---|---|---|---|---|---|---|---|---|---|---|
| | | | Core | Position | Conservancy | Servers | Number of Alleles | Alleles | Position | Core | Alleles | Number of Alleles | Conservancy | Servers | | | | |
| Nucleocapsid (N) | 1 | RGGSQASSRSSSRSR | | 177–191 | 99.46% (184/185) | 2 | 1 | HLA-DQA1*05:01/DQB1*03:01 | 178–192 | | HLA-DQA1*05:01/DQB1*03:01 | 1 | 79.73% (240/301) | 2 | Antigen (1.0041) | Non-allergen | Positive | |
| | 2 | LALLLDRLNQLESK | - | 219–233 | 100.00% (185/185) | 2 | 19 | HLA-DRB1*03:01, HLA-DRB3*01:01, HLA-DRB4*01:01, HLA-DRB1*12:01, HLA-DPA1*03:01/DPB1*04:02, HLA-DQA1*01:01/DQB1*05:01, HLA-DRB1*13:02, HLA-DPA1*02:01/DPB1*01:01, HLA-DPA1*02:01/DPB1*05:01, HLA-DPA1*01:03/DPB1*04:01, HLA-DRB1*04:05, HLA-DPA1*01:03/DPB1*02:01, HLA-DRB1*04:01, HLA-DRB1*11:01, HLA-DRB3*02:02, HLA-DRB1*08:02, HLA-DRB1*15:01, HLA-DRB1*01:01, HLA-DRB5*01:01 | 220–234 | - | HLA-DRB1*03:01, HLA-DRB3*01:01, HLA-DRB4*01:01, HLA-DRB1*12:01, HLA-DPA1*03:01/DPB1*04:02, HLA-DQA1*01:01/DQB1*05:01, HLA-DRB1*13:02, HLA-DPA1*02:01/DPB1*01:01, HLA-DPA1*02:01/DPB1*05:01, HLA-DPA1*01:03/DPB1*04:01, HLA-DRB1*04:05, HLA-DPA1*01:03/DPB1*02:01, HLA-DRB1*04:01, HLA-DRB1*11:01, HLA-DRB3*02:02, HLA-DRB1*08:02, HLA-DRB1*15:01, HLA-DRB1*01:01, HLA-DRB5*01:01 | 19 | 84.72% (255/301) | 2 | Antigen (0.7357) | Non-allergen | Positive | [77] |
| | 3 | YKHWPQIAQFAPSAS | - | 298–312 | 100.00% (185/185) | 2 | 6 | HLA-DQA1*05:01/DQB1*03:01, H, LA-DQA1*01:02/DQB1*06:02, HLA-DRB1*09:01, HLA-DRB1*01:01, HLA-DRB1*04:01, HLA-DRB1*04:05 | 299–313 | - | HLA-DQA1*05:01/DQB1*03:01, HLA-DQA1*01:02/DQB1*06:02, HLA-DRB1*09:01, HLA-DRB1*01:01, HLA-DRB1*04:01, HLA-DRB1*04:05 | 6 | 82.72% (249/301) | 2 | Antigen (0.434) | Non-allergen | Positive | [77] |
| | 4 | ASAFFGMSRIGMEVT | FFGMSRIGM | 311–325 | 100.00% (185/185) | 1 & 2 | 18 | HLA-DRB1*11:01, HLA-DRB1*04:05, HLA-DRB1*01:01, HLA-DRB1*08:02, HLA-DRB1*09:01, HLA-DQA1*01:02/DQB1*06:02, HLA-DRB5*01:01, HLA-DPA1*01:03/DPB1*04:01, HLA-DRB1*07:01, HLA-DQA1*05:01/DQB1*03:01, HLA-DPA1*04:01/HLA-DPA1*01:03/DPB1*02:01, HLA-DRB1*15:01, HLA-DPA1*02:01/DPB1*01:01, HLA-DPA1*03:01/DPB1*04:02, HLA-DRB1*12:01, HLA-DRB4*01:01, HLA-DRB1*13:02 | 312–326 | FFGMSRIGM | HLA-DRB1*11:01, HLA-DRB1*04:05, HLA-DRB1*01:01, HLA-DRB1*08:02, HLA-DRB1*09:01, HLA-DQA1*01:02/DQB1*06:02, HLA-DRB5*01:01, HLA-DPA1*01:03/DPB1*04:01, HLA-DRB1*07:01, HLA-DQA1*05:01/DQB1*03:01, HLA-DPA1*04:01/HLA-DPA1*01:03/DPB1*02:01, HLA-DRB1*15:01, HLA-DPA1*02:01/DPB1*01:01, HLA-DPA1*03:01/DPB1*04:02, HLA-DRB1*12:01, HLA-DRB4*01:01, HLA-DRB1*13:02 | 18 | 84.72% (255/301) | 1 & 2 | Antigen (0.8620) | Non-allergen | Negative | [77] |
| | 5 | MSRIGMEVTPSGTWL | - | 317–331 | 100.00% (185/185) | 2 | 7 | HLA-DRB1*04:01, HLA-DRB1*04:05, HLA-DQA1*05:01/DQB1*03:01, HLA-DRB1*08:02, HLA-DRB4*01:01, HLA-DRB1*01:01, HLA-DRB1*09:01 | 318–332 | - | HLA-DRB1*04:01, HLA-DRB1*04:05, HLA-DQA1*05:01/DQB1*03:01, HLA-DRB1*08:02, HLA-DRB4*01:01, HLA-DRB1*01:01, HLA-DRB1*09:01 | 7 | 83.06% (250/301) | 2 | Antigen (0.7397) | Non-allergen | Negative | [59] |

(*Continued*)

**Table 9.** (Continued)

| SARS-CoV-2 protein | Sl. No. | Peptide | SARS-CoV-2 | | | | | | SARS-CoV | | | | | | Probable Antigenicity by Vexijen (Threshold = 0.4) | Probable Allergenicity by AllerTop | IFN gamma epitope | Experimentally verified epitopes / VIPR reference |
|---|---|---|---|---|---|---|---|---|---|---|---|---|---|---|---|---|---|---|
| | | | Core | Position | Alleles | Number of Alleles | Conservancy | Servers | Core | Position | Alleles | Number of Alleles | Conservancy | Servers | | | | |
| Membrane (M) | 1 | PVTLACFVLAAVYRI | - | 59–73 | HLA-DPA1*01:03/DPB1*02:01, HLA-DPA1*01:03/DPB1*04:01, HLA-DPA1*03:01/DPB1*04:02, HLA-DQA1*01:02/DQB1*06:02, HLA-DQA1*01:01/DQB1*05:01, HLA-DRB1*09:01, HLA-DRB1*01:01, HLA-DRB1*15:01, HLA-DPA1*02:01/DPB1*01:01, HLA-DRB1*12:01, HLA-DRB3*01:01, HLA-DRB5*01:01, HLA-DRB1*04:05, HLA-DRB1*13:02, HLA-DRB1*11:01, HLA-DQA1*05:01/DQB1*03:01, HLA-DRB1*08:02, HLA-DRB1*04:01, HLA-DRB4*01:01, HLA-DRB1*07:01 | 20 | 98.24% (167/170) | 2 | - | 58–72 | HLA-DRB1*07:01, HLA-DPA1*01:03/DPB1*02:01, HLA-DPA1*01:03/DPB1*04:01, HLA-DPA1*01:02/DQB1*06:02, HLA-DQA1*01:01/DQB1*05:01, HLA-DRB1*09:01, HLA-DRB1*01:01, HLA-DRB1*15:01, HLA-DPA1*02:01/DPB1*01:01, HLA-DRB1*12:01, HLA-DRB3*01:01, HLA-DRB5*01:01, HLA-DRB1*04:05, HLA-DRB1*13:02, HLA-DRB1*11:01, HLA-DQA1*05:01/DQB1*03:01, HLA-DRB1*08:02, HLA-DRB1*04:01, HLA-DRB4*01:01 | 20 | 78.73% (211/268) | 2 | Antigen (0.8548) | Non-allergen | Positive | - |
| | 2 | PKEITVATSRTLSYY | ITVATSRTL | 165–179 | HLA-DRB1*01:02, HLA-DRB1*09:01, HLA-DRB1*07:01, HLA-DRB1*09:01, HLA-DRB1*08:02, HLA-DRB1*13:02, HLA-DRB1*03:01, HLA-DRB1*01:01, HLA-DRB1*15:01, HLA-DRB3*02:02, HLA-DRB1*12:01, HLA-DRB5*01:01, HLA-DQA1*01:02/DQB1*06:02, HLA-DRB1*04:01, HLA-DRB1*11:01, HLA-DRB4*01:01, HLA-DQA1*05:01/DQB1*03:01, HLA-DRB1*04:05 | 18 | 98.24% (167/170) | 1 & 2 | ITVATSRTL | 164–178 | HLA-DRB1*07:01, HLA-DRB1*09:01, HLA-DRB1*08:02, HLA-DRB1*13:02, HLA-DRB1*03:01, HLA-DRB1*01:01, HLA-DRB1*15:01, HLA-DRB3*02:02, HLA-DRB1*12:01, HLA-DRB5*01:01, HLA-DQA1*01:02/DQB1*06:02, HLA-DRB1*04:01, HLA-DRB1*11:01, HLA-DRB4*01:01, HLA-DQA1*05:01/DQB1*04:05 | 16 | 79.48% (213/268) | 1 & 2 | Antigen (0.5800) | Non-allergen | Negative | |

(Continued)

**Table 9.** (Continued)

| SARS-CoV-2 protein | Sl. No. | Peptide | SARS-CoV-2 | | | | | | SARS-CoV | | | | | | Probable Antigenicity by Vexifen (Threshold = 0.4) | Probable Allergenicity by AllerTop | IFN gamma epitope | Experimentally verified epitopes / ViPR reference |
|---|---|---|---|---|---|---|---|---|---|---|---|---|---|---|---|---|---|---|
| | | | Core | Position | Alleles | Number of Alleles | Conservancy | Servers | Core | Position | Alleles | Number of Alleles | Conservancy | Servers | | | | |
| Envelope (E) | 1 | VLLFLAFVVFLLVTL | - | 17–31 | HLA-DPA1*01:03/DPB1*04:01, HLA-DPA1*01:03/DPB1*02:01 | 2 | 100.00% (177/177) | 2 | - | 17–31 | HLA-DPA1*01:03/DPB1*04:01, HLA-DPA1*01:03/DPB1*02:01 | 2 | 82.86% (203/245) | 2 | Antigen (0.6386) | Non-allergen | Positive | [59] |
| | 2 | AFVVFLLVTLAILTA | - | 22–36 | HLA-DPA1*01:03/DPB1*02:01, HLA-DRB1*01:01 | 2 | 100.00% (177/177) | 2 | - | 22–36 | HLA-DPA1*01:03/DPB1*02:01, HLA-DRB1*01:01 | 2 | 86.12% (211/245) | 2 | Antigen (0.6229) | Non-allergen | Positive | [59] |
| | 3 | LVTLAILTALRLCAY | - | 28–42 | HLA-DRB1*12:01, HLA-DRB1*15:01, HLA-DRB4*01:01, HLA-DRB1*01:01, HLA-DRB1*08:02, HLA-DRB1*11:01, HLA-DPA1*02:01/DPB1*05:01, HLA-DRB5*01:01, HLA-DPA1*03:01/DPB1*04:02, HLA-DRB1*04:05, HLA-DPA1*01:03/DPB1*04:01, HLA-DQA1*01:02/DQB1*06:02, HLA-DRB1*07:01, HLA-DPA1*01:03/DPB1*02:01, HLA-DPA1*02:01/DPB1*01:01, HLA-DRB1*03:01, HLA-DRB1*04:01, HLA-DRB1*13:02, HLA-DRB1*09:01 | 19 | 99.44% (176/177) | 2 | - | 28–42 | HLA-DRB1*12:01, HLA-DRB1*15:01, HLA-DRB4*01:01, HLA-DRB1*01:01, HLA-DRB1*08:02, HLA-DRB1*11:01, HLA-DPA1*02:01/DPB1*05:01, HLA-DRB5*01:01, HLA-DPA1*03:01/DPB1*04:02, HLA-DRB1*04:05, HLA-DPA1*01:03/DPB1*04:01, HLA-DQA1*01:02/DQB1*06:02, HLA-DRB1*07:01, HLA-DPA1*01:03/DPB1*02:01, HLA-DPA1*02:01/DPB1*01:01, HLA-DRB1*03:01, HLA-DRB1*04:01, HLA-DRB1*13:02, HLA-DRB1*09:01 | 19 | 82.45% (202/245) | 2 | Antigen (0.407) | Non-allergen | Negative | [59] |
| | 4 | LCAYCCNIVNVSLVK | - | 39–53 | HLA-DRB3*02:02, HLA-DRB1*13:02, HLA-DRB1*04:01, HLA-DRB1*04:05, HLA-DRB1*07:01, HLA-DRB1*01:01, HLA-DRB1*08:02, HLA-DRB1*09:01, HLA-DRB1*15:01, HLA-DRB4*01:01 | 10 | 100.00% (177/177) | 2 | - | 39–53 | HLA-DRB3*02:02, HLA-DRB1*13:02, HLA-DRB1*04:01, HLA-DRB1*04:05, HLA-DRB1*07:01, HLA-DRB1*01:01, HLA-DRB1*08:02, HLA-DRB1*09:01, HLA-DRB1*15:01, HLA-DRB4*01:01 | 10 | 84.08% (206/245) | 2 | Antigen (0.8552) | Non-allergen | Negative | - |

CD4+ T-cell epitopes were identified using two independent prediction algorithms and servers, which are numbered as (1) for NetMHCpan4 and (2) for TepiTool.

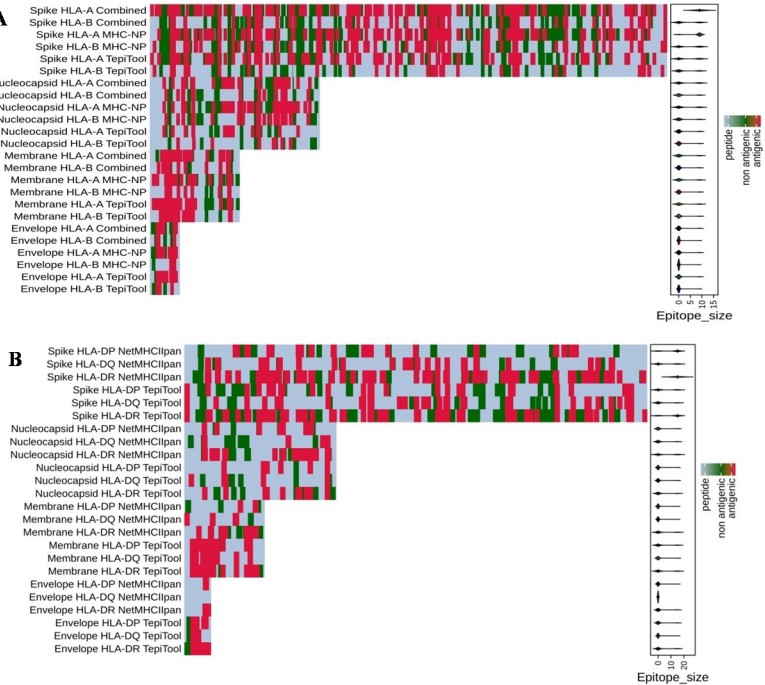

**Fig 5. Summary of SARS-CoV-2-derived T-cell epitopes.** Heat map showing the distribution of **(a)** HLA class I and **(b)** HLA class II epitopes across the protein sequences of spike (1273 aa), nucleocapsid (419 aa), membrane (222 aa), and envelope (75 aa) proteins of SARS-CoV-2. Red color represents antigenic epitopes predicted using the methods described in Fig 1.

## Cross-validation of predicted epitopes of SARS-CoV-2 in the NIAID Virus Pathogen Database and Analysis Resource and published literature

The Virus Pathogen Database and Analysis Resource (ViPR) database funded by the National Institute of Allergy and Infectious Diseases (NIAID), USA, provides two types of immune-related information on antigens; the predicted epitopes (NetCTL 1.2 server) and the experimentally determined epitopes derived from the IEDB. The B- and T-cell epitopes predicted in this study were searched on the ViPR database (https://www.viprbrc.org) by selecting different parameters such as family-*Coronaviridae*, human host, and experimentally determined B- and T-cell epitopes. Two different assays, positive T-cell and positive MHC-binding assays were applied for testing the T-cell response against epitopes and epitope-MHC binding, respectively [67]. We have furnished the unique identification number (IEDB ID) of some predicted B- and T-cell epitopes of structural proteins of SARS-CoV-2, which are identical to experimentally determined epitopes of structural proteins of SARS-CoV (Tables 2–10). It is pertinent to note that the predicted continuous B-cell epitopes reported in this study have been previously identified using similar computational tools, and some of the predicted epitopes have now been experimentally validated (Tables 2–9). Similar findings presented in this work have been reported previously [27–38].

## Identification of potential B- and T-cell epitopes of structural proteins for development of serological assays and multi-epitope-based vaccines

Generally, the S and N proteins are the major targets for the development of vaccines and diagnostic tools against SARS-CoV-2 using recombinant antigens. The development of rapid

**Table 10. List of SARS-CoV-2 proteins that were predicted as common and overlapping B- and T- cell epitopes.**

| SARS-CoV-2 protein | Name of epitope | Predicted epitope of SARS-CoV-2 | Length | Antigenicity by Vexijen (T = 0.4) | Allergenicity by AllerTope v. 2.0 | Agadir Score | Conservancy analysis by IEDB tool | | Experimentally verified epitopes / ViPR reference |
|---|---|---|---|---|---|---|---|---|---|
| | | | | | | | SARS-CoV-2 | SARS-CoV | |
| Spike (S) | Linear and conformational B-cell epitopes | 1253-CCKFDEDDSEPVLKGVKLHYT-1273 | 21 | Antigen (0.9101) | Non-allergen | 0.26 | 94.86% (166/175) | 83.74% (273/326) | (IEDB-ID: 6476) [59] |
| | Linear B-cell epitopes and MHC I | 403-RGDEVRQIAPGQTGKIADYNYKLPD-427 | 25 | Antigen (1.0356) | Non-allergen | 0.45 | 100.00% (175/175) | 0.00% (0/326) | - |
| | | 437-NSNNLDSKVGGNYNYLYRLFRKSNL-461 | 25 | Antigen (0.4015) | Non-allergen | 4.42 | 100.00% (175/175) | 0.00% (0/326) | (MHC I, IEDB-ID: 1074979) [59] |
| Nucleocapsid (N) | Linear and conformational B-cell epitopes and MHC I | 366-TEPKKDKKKKADETQALPQRQKKQQTVTL-394 | 29 | Antigen (0.5248) | Non-allergen | 0.63 | 100.00% (185/185) | 0.00% (0/301) | - |
| | Linear B-cell epitopes and MHC I | 173-AEGSRGGSQASSRSSSRS-190 | 18 | Antigen (0.8081) | Non-allergen | 0.21 | 99.46% (184/185) | 79.73% (240/301) | (B-cell, IEDB-ID: 22481) [59] |
| | | 227-LNQLESKMSGKGQQQQGQTVTKKSAAEASKKPRQKRTATK-266 | 40 | Antigen (0.5387) | Non-allergen | 1.9 | 100.00% (185/185) | 0.00% (0/301) | (B-cell, IEDB-ID: 31692) [59] |
| Membrane (M) | Linear and conformational B-cell epitopes and MHC I | 160-DIKDLPKEITVATSRTLSYYKLG-182 | 23 | Antigen (0.7442) | Non-allergen | 0.31 | 98.24% (167/170) | 79.48% (213/268) | (B-cell, IEDB-ID: 48052) [59] |
| Envelope (E) | Linear and conformational B-cell epitopes | 57-YVYSRVKNLNSSRVPDLL-75 | 19 | Antigen (0.565) | Non-allergen | 0.2 | 100.00% (177/177) | 0.00% (0/245) | - |

**Table 11. Design of multi-epitope vaccine constructs of SARS-CoV-2.**

| Multi-epitope constructs of SARS-CoV-2 | Protein | Linear B-cell epitope | Conformational B-cell epitope | CTL epitope | HTL epitope |
|---|---|---|---|---|---|
| RBD of spike protein (B- and T-cell epitopes) | Spike | 331-NITNLCPFGEVFNATRFASVYAWNRK-356, 370-NSASFSTFKCYGVSPTKLNDLCFTNV-395, 403-RGDEVRQIAPGQTGKIADYNYKLPD-427, 437-NSNNLDSKVGGNYNYLYRLFRKSNI-461, 525-CGPKKSTNLVKNKCVNFNFNGL-546 | 304-KSFTVEKGIYQTSNFRVQPTES-325, 390-LCFTNVYADSFVIR-403, 530-STNLVKNKNTESNKFLPFQQAVRDPQTLEIL-585 | 409-QIAPGQTGK-417, 443-SKVGGNYNY-451, 505-YQPYRVVVL-513, 512-VLSFELLHA-520 | 310-SASAFFGMSRIGMEV-324, 374-FSTFKCYGVSPTKLN-388, 512-VLSFELLHAPATVCG-520 |
| Spike protein (B- and T-cell epitopes) | Spike | 370-NSASFSTFKCYGVSPTKLNDLCFTNV-395, 403-RGDEVRQIAPGQTGKIADYNYKLPD-427, 437-NSNNLDSKVGGNYNYLYRLFRKSNI-461, 770-IAVEQDKNTQEVFAQVKQ-787, 1028-KMSECVLGQSKRVDFCGKGYHL-1049, 1191-KNLNESLIDLQELGKYEQYIK-1211, 1253-CCKFDEDDSEPVLKGVKLHYT-1273 | 175-FLMDLEGKQGNFKNLR-190, 904-YRFNGIGVTQNVLYENQK-921, 1161-SPDVDLGDISGINASVVN-1178, 1204-GKYEQYIKWPWYIWLGFI-1221, 1254-CKFDEDDSEPVLKGVKLH-1271 | 507-PYRVVVLSF-515, 590-CSFGGVSVI-598, 886-WTFGAGAAL-894, 898-FAMQMAYRF-909, 1223-GLIAIVMVT-1231 | 539-VNFNFNGLTGTGVLT-553, 753-LLQYGSFCTQLNRAL-767, 819-EDLLFNKVTLADAGF-833, 894-LQIPFAMQMAYRFNG-908, 906-FNGIGVTQNVLYENQ-920, 1014-RAAEIRASANLAATK-1028, 1038-KRVDFCGKGYHLMSF-1052 |
| Structural protein construct (B- and T-cell epitopes of S, N, M and E protein) | Spike | 370-NSASFSTFKCYGVSPTKLNDLCFTNV-395, 403-RGDEVRQIAPGQTGKIADYNYKLPD-427, 437-NSNNLDSKVGGNYNYLYRLFRKSNI-461, 770-IAVEQDKNTQEVFAQVKQ-787, 1028-KMSECVLGQSKRVDFCGKGYHL-1049, 1191-KNLNESLIDLQELGKYEQYIK-1211, 1253-CCKFDEDDSEPVLKGVKLHYT-1273 | 175-FLMDLEGKQGNFKNLR-190, 904-YRFNGIGVTQNVLYENQK-921, 1161-SPDVDLGDISGINASVVN-1178, 1204-GKYEQYIKWPWYIWLGFI-1221, 1254-CKFDEDDSEPVLKGVKLH-1271 | 507-PYRVVVLSF-515, 590-CSFGGVSVI-598, 886-WTFGAGAAL-894, 898-FAMQMAYRF-909, 1223-GLIAIVMVT-1231 | 539-VNFNFNGLTGTGVLT-553, 753-LLQYGSFCTQLNRAL-767, 819-EDLLFNKVTLADAGF-833, 894-LQIPFAMQMAYRFNG-908, 906-FNGIGVTQNVLYENQ-920, 1014-RAAEIRASANLAATK-1028, 1038-KRVDFCGKGYHLMSF-1052 |
| | Nucleocapsid | 32- RSGARSKQRRPQGLP-46,173- AEGSRGGSQASSRSSSRS-190,227- LNQLESKMSGKGQQQQGTVTKKSAAEASKKPRQKRTATK- 266,366- TEPKKDKKKKADETQALPQRQKKQQTVTL-394 | 95-RGGDGKMKDLSPRWYFYYLG-114, 104-LSPRWYFYYLGTGP-117, 272-QAFGRRGPEQTQGNFGDQ-289 | 104-LSPRWYFYY-112, 177-RGGSQASSR-185, 306-QFAPSASAF-314, 316-GMSRIGMEV-324, 323-EVTPSGTWL-331 | 177-RGGSQASSRSSSRSR-191, 219-LAILLLDRLNQLESK-233, 298-YKHWPQIAQFAPSAS-312, 303-QIAQFAPSASAFFGM-317, 311-ASAFFGMSRIGMEVT-325, 317-MSRIGMEVTPSGTWL-331 |
| | Membrane | 101-RLFARTRSMWSFNPETNIL-119, 160-DIKDLPKEITVATSRTLSYYKLG-182 | 48-IIKLIFIWLLWPVTLACFVLAAVR-72, 176-LSYYKLGASQRV-187, 189-GDFAASYRIGNYKLNTDH-210 | 6-GTITVEELK-14, 54-LWLLWPVTL-62, 198-RYRIGNYKL-206, 135-ESELVIGAV-143 | 59-PVTLACFVLAAVYRI-73, 165-PKEITVATSRTLSYI-179 |
| | Envelope | 57-YVYSRVKNLNSSRVPDLLV-75 | 1-MYSFVSEETGTLIVNSVLLFLAFVVFLLVTL-31, 56-FYVYSRVKNLNSSRVPDLLV-75 | 16-SVLLFLAFV-24, 20-FLAFVVFLL-28, 26-FLLVTLAIL-34, 38-RLCAYCCNI-46 | 17-VLLFLAFVVFLLVTL-31, 28-LVTLAILTALRLCAY-42, 39-LCAYCCNIVNVSLVK-53 |

(Continued)

**Table 11.** (Continued)

| Multi-epitope constructs of SARS-CoV-2 | Protein | Linear B-cell epitope | Conformational B-cell epitope | CTL epitope | HTL epitope |
|---|---|---|---|---|---|
| Chimeric construct of S and N proteins (B- and T-cell epitopes of S and N) | Spike | 370-NSASFSTFKCYGVSPTKLNDLCFTNV-395, 403-RGDEVRQIAPGQTGKIADYNYKLPD-427, 437-NSNNLDSKVGGNYNYLYRLFRKSNL-461, 770-IAVEQDKNTQEVFAQVKQ-787, 1028-KMSECVLGQSKRVDFCGKGYHL-1049, 1191-KNLNESLIDLQELGKYEQYIK-1211, 1253-CCKFDEDDSEPVLKGVKLHYT-1273 | 175-FLMDLEGKQGNFKNLR-190, 904-YRFNGIGVTQNVLYENQK-921, 1161-SPDVDLGDISGINASVVN-1178, 1204-GKYEQYIKWPWYIWLGFI-1221, 1254-CKFDEDDSEPVLKGVKLH-1271 | 507-PYRVVVLSF-515, 590-CSFGGVSVI-598, 886-WTFGAGAAL-894, 898-FAMQMAYRF-909, 1223-GLIAIVMVT-1231 | 539-VNFNFNGLIGTGVLT-553, 753-LLQYGSFCTQLNRAL-767, 819-EDLLFNKVTLADAGF-833, 894-LQIFFAMQMAYRFNG-908, 906-FNGIGVTQNVLYENQ-920, 1014-RAAEIRASANLAATK-1028, 1038-KRVDFCGKGYHLMSF-1052 |
| | Nucleocapsid | 32- RSGARSKQRRPQGLP-46, 173-AEGSRGGSQASSRSSRS-190, 227-LNQLESKMSGKGQQQQGQTVTKKSAAEASKKPRQKRTATK-266, 366- TEPKKDKKKKADETQALPQRQKKQQTVTL-394 | 95-RGGDGKMKDLSPRWYFYYLG-114, 104-LSPRWYFYYLGTGP-117, 272-QAFGRRGPEQTQGNFGDQ-289 | 104-LSPRWYFYY-112, 177-RGGSQASSR-185, 306-QFAPSASAF-314, 316-GMSRIGMEV-324, 323-EVTPSGTWL-331 | 177-RGGSQASSRSSSRSR-191, 219-LALLLLDRLNQLESK-233, 298-YKHWPQIAQFAPSAS-312, 303-QIAQFAPSASAFFGM-317, 311-ASAFFGMSRIGMEVT-325, 317-MSRIGMEVTPSGTWL-331 |

List of potent B- and T-cell epitopes that are included in the multi-epitope vaccine constructs.

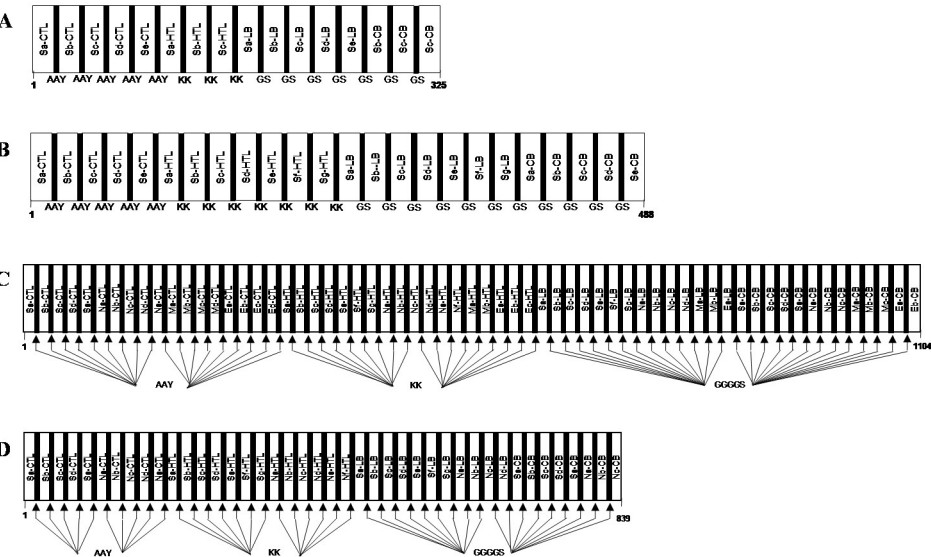

**Fig 6. Schematic diagram of multi-epitope vaccine constructs of SARS-CoV-2.** The multi-epitope vaccine constructs consisted of top-ranked predicted and experimentally validated B- and T-cell epitopes. Cytotoxic T lymphocytes (CTL) epitopes were joined by the AAY linker, whereas helper T lymphocytes (HTL) and B-cell epitopes were joined by KK and GGGGS linkers, respectively. **(a)** The receptor-binding domain (RBD) of the spike (S) protein (B- and T-cell epitopes); **(b)** The S protein (B- and T-cell epitopes), **(c)** The structural protein construct (B- and T-cell epitopes of S, N, M, and E proteins); **(d)** The chimeric construct of S and N proteins (B- and T-cell epitopes of S and N proteins).

antibody tests requires the production of recombinant antigens and their validation in an enzyme-linked immunosorbent assay (ELISA) using the convalescent serum of patients with COVID-19. In addition, ELISA does not require the culture of SARS-CoV-2 in a BSL-3 containment facility [10–14, 68–70]. In this study, we designed a multi-epitope chimeric construct, using the computationally predicted B- and T-cell epitopes of the S, E, M, and N proteins of SARS-CoV-2 (Table 6). In the multi-epitope vaccine constructs, we included MHC class I (CTL)- and class II (HTL)-binding antigenic, non-allergenic, and conserved epitopes, which were predicted to elicit IFN-γ release. The CTL and HTL epitopes were linked together by AAY and KK cleavable linkers, whereas the B-cell epitopes (linear and conformational) were linked together with the GGGGS flexible linker (Fig 6a–6d). The following four multi-epitope chimeric vaccine constructs containing N-CTL, HTL and B-cell epitopes were designed: (i) The RBD of the S protein (B- and T-cell epitopes), (ii) the full-length S protein (B- and T-cell epitopes), (iii) the structural protein construct (B- and T-cell epitopes of S, E, M, and N proteins), and (iv) the chimeric construct of S and N proteins (B- and T-cell epitopes of S and N proteins) (Fig 6).

The structure of multi-epitope vaccine constructs was modeled using I-TASSER and was validated by the RAMPAGE server to generate a Ramachandran plot [71] (S1a-S1d Fig in S1 File). Most of the amino acid residues of epitope constructs were found in the favorable region (S3 Fig in S1 File, Inset table). Various physicochemical parameters including the number of residues, theoretical pI, molecular weight, aliphatic index, and grand average of hydrophobicity (GRAVY), were analysed using ProtParam [48]. Based on the aliphatic index scores, the multi-epitope constructs might be considered moderately thermostable (S1 Table in S1 File). GRAVY scores obtained displayed negative value for all the constructs, indicating the likelihood of the chimeric multi-epitope constructs being globular and hydrophilic in nature. The secondary structure of multi-epitope constructs was predicted using the online server

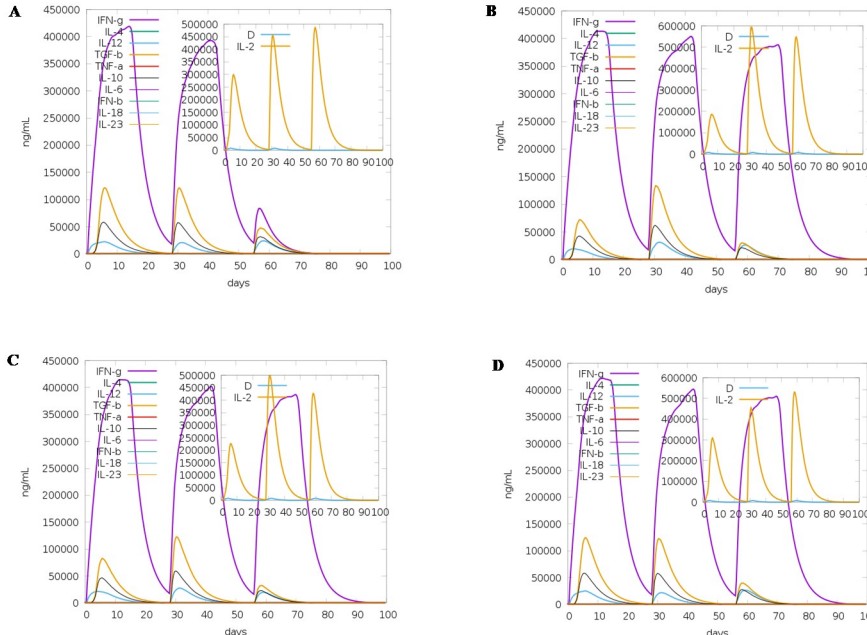

**Fig 7.** *In silico* **immune simulation of multi-epitope vaccine constructs of SARS-CoV-2.** C-ImmSim simulation of the cytokine levels induced by three injections administered four weeks apart. The main plot shows cytokine levels after the injections. The inset plot shows IL-2 levels with the Simpson index (D), which is indicated by the dotted line, is a measure of diversity. Increase in D over time indicates the emergence of different epitope-specific dominant clones of T cells. The smaller the D value, the lower the diversity. **(a)** The receptor-binding domain (RBD) of the spike (S) protein (B- and T-cell epitopes); **(b)** The S protein (B- and T-cell epitopes); **(c)** The structural protein construct (B- and T-cell epitopes of S, N, M, and E proteins); **(d)** The chimeric construct of S and N proteins (B- and T-cell epitopes of S and N proteins).

PSIPRED [51], and the alpha-helical content of the constructs is provided in S4a-S4d Fig in S1 File.

The cleavable linkers are required to be accessible for the proteases associated with the MHC I and II antigen processing pathways [49]. We observed that the cleavable linkers (AAY and KK) included in multi-epitope vaccines were predicted on the accessible region of vaccine constructs, indicating a high probability of T-cell epitope presentation by MHC molecules (S2a-S2d in S1 File). The results of C-ImmSim server (http://150.146.2.1/C-IMMSIM/index.php) prediction revealed the multi-epitope constructs were able to stimulate cytokine production, including IFN-γ, following immunization with the peptide (Fig 7a–7d). *In silico* immune simulation of the multi-epitope constructs {(a) RBD of the S protein, (b) the S protein, (c) structural protein construct, and (d) chimeric construct of S and N proteins)} showed consistent correlation with the actual immune responses as observed by the primary response of high levels of IgM. This is followed by a marked increase in B-cell populations and levels of $IgG_1$ + $IgG_2$, IgM, and IgG + IgM antibodies, as a part of the secondary and tertiary responses S5a-S5d in S1 File [i-iv]). A similarly high response was seen in the $T_H$ (helper) and $T_C$ (cytotoxic) cell populations with corresponding memory development, especially for constructs made of structural proteins (S, E, M, and N) and chimeric constructs of S and N proteins (S5c and S5d Fig in S1 File).

## Discussion

Research communities around the world are in the process of developing safe, effective, and affordable vaccines against COVID-19. As of June 3, 2021, the World Health Organization has

evaluated and recommended the emergency use of COVID-19 vaccines developed by AstraZeneca/Oxford, Moderna, Johnson and Johnson, Pfizer/BioNTech, Sinopharm, and Sinovac Biotech based on satisfactory published findings on the safety and efficacy profiles of these vaccines [19–26, 72, 73]. Although data on immunological characteristics of patients with COVID-19 are limited, growing experimental evidence suggests that B and T cells are naturally activated during infection, and they contribute substantially towards the resolution of SARS-CoV-2 infection [16–18, 74, 75]. Importantly, SARS-CoV-2-specific antibodies which are found in convalescent sera of approximately 95% of patients with COVID-19 can neutralize the virus, and the antibody titers correlate positively with measured levels of immunoglobulins that mainly target S and N proteins [7–9, 14, 76]. There have been interesting observations on virus-specific CD4+ and CD8+ T cells in sera from both patients with acute COVID-19 and convalescent patients. A similar development of T cells in SARS-CoV-2-unexposed healthy individuals might have important implications for new design and analysis of ongoing vaccine trials [16–18]. A proportion of the SARS-CoV-2-specific CD8+ T cells isolated from convalescent peripheral blood of patients with COVID-19 exhibited the undesirable "exhausted" phenotype; such perturbations of T-cell subsets may eventually weaken the antiviral immunity of the host [77]. While efforts to develop vaccines that are more effective are ongoing, there is a need to develop a better understanding of B- and T-cell responses against SARS-CoV-2.

Of the linear B-cell epitopes predicted in the RBD of the S protein (Fig 4, Inset table), the synthetic peptide spanning the regions aa331-356, aa370-395, aa437-461, and aa483-493 has been shown to react with convalescent sera from patients with COVID-19 [61, 77]. Intriguingly, most of the amino acid residues spanning the predicted B-cell epitope (aa331-356 and aa437-461, Fig 4, Inset table) of the S protein have been shown to interact with the cross-neutralizing mAb S309 in an ACE2 receptor-independent manner [10, 62]. In a related study using the recombinant SARS-CoV-2 RBD antigen, a strong correlation between levels of RBD-binding antibodies and SARS-CoV-2-neutralizing antibodies was observed in patients with COVID-19 [11, 78]. Similar investigations performed using samples collected from patients with COVID-19 in the USA, Europe, and Hong Kong have detected specific and sensitive antibodies using the full-length and RBD of the S protein [69, 78]. Interestingly, the predicted linear B-cell epitope (aa653-666, Table 2) has now been experimentally validated using the synthetic peptide aa655-672 of the S protein, which was abundantly detected in samples derived from patients with COVID-19 [79].

The S and N proteins are considered the main targets for the development of vaccines and immunoassays against COVID-19. Antibodies to N protein appear earlier than S antibodies are found more sensitive for detection of early infection of SARS-CoV-2 [14]. Leung *et al.* (2004) showed that antibody responses specific to the N protein were detected during early infection with SARS-CoV [5]. The B-cell epitope predicted and identified (aa359-403) in the N protein of SARS-CoV was previously shown to react abundantly with the serum of patients with SARS-CoV [80, 81]. The C-terminal epitope (`aa366-TEPKKDKKKKADET-QALPQRQKKQQTVTL-394`) predicted in the N protein of SARS-CoV-2 was located on the accessible region of the virus protein structure; however, the immunogenicity of this predicted epitope requires further experimental validation. A recent study used a peptide-based SARS-CoV-2 proteome microarray and identified several B-cell peptides (approximately 5-mer in length) in the serum of patients with COVID-19 [82]. Many of these peptides partially overlap with the predicted B-cell epitopes of S, N, and M proteins presented in this study (Table 2). Previously, 206 monoclonal antibodies specific to the RBD of the S protein of SARS-CoV-2 have been characterized in infected individuals [83]. Wang *et al.* (2020) identified a human monoclonal antibody (47D11) that neutralizes SARS-CoV-2 and SARS-CoV by binding to their respective RBDs without hampering ACE2 receptor interaction [12]. Amanat *et al.*

(2020) developed a serological assay using the S protein expressed in insect and mammalian expression systems; the results indicated plasma/serum samples derived from patients with COVID-19 reacted strongly to both the RBD and the full-length S protein [68]. To date, the findings from published work juxtaposed with those of our study might have significant implications in the rational designing of epitope-based vaccines and serological diagnostic assays based on S and N proteins of SARS-CoV-2 [10–14, 68, 70].

Increasing experimental evidence suggests the development of CD4+ and CD8+ T-cell responses in a majority of patients who recovered from COVID-19; however, their role in disease progression and protection is not well understood [16–18, 65]. For example, the presence of a high proportion of CD4+ T cells in SARS-CoV-2-unexposed healthy individuals might be attributed to cross-reactivity of the C-terminus of the S protein, which shares homology with the C-terminus of spike glycoproteins in human endemic coronaviruses [18]. The identification of non-spike dominant CD8+ T-cell epitopes suggest that other proteins such as E, N, M, and ORFs can be used to design effective vaccines [17]. Therefore, in this study, we considered the four structural proteins (S, E, M, and N) for the mapping of T-cell epitopes based on data from published literature regarding immune responses to SARS-CoV and SARS-CoV-2 [16–18]. We selected the T-cell epitopes based on predicted parameters such as antigenicity, conservancy, non-allergenicity, and ability to elicit IFN-γ response. The CD8+ T-cell epitope of S protein (aa268-GYLQPRTFL-276, Table 6) predicted in this study has been experimentally verified to induce IFN-γ response in peripheral blood monocular cells (PBMCs) isolated from patients who recovered from COVID-19 [17, 84]. Similarly, the CD8+ T-cell epitopes predicted for the S protein (aa505-520, aa755-765, aa873-881, and aa1209-1217), N protein (aa66-74, aa102-110, and aa343-369), and M protein (aa54-62, aa64-72, and aa172-180) have been experimentally validated using PBMCs of patients with COVID-19 (Table 6) [17, 67, 85]. Notably, the highly conserved CD8+ T-cell epitope (aa361-369) of the N protein (Table 6) has been detected in convalescent sera from patients with COVID-19 [17]. Therefore, these findings from previous studies endorse that epitope predicted through validated computational tools can help to rapidly identify immunogenic epitopes. The presence of conserved CD4+ T-cell epitopes common to both SARS-CoV and SARS-CoV-2 in proteins such as S (aa 1014–1028, aa539-553, aa899-913), N (aa219-233, aa298-312, aa311-325, aa317-331), and E (aa17-31) have already been validated experimentally [11, 80]. We have provided the unique IEDB IDs of predicted B- and T-cell epitopes of SARS-CoV-2 that matched with experimentally validated epitopes of SARS-CoV available in the ViPR database. The S, M, and N proteins of SARS-CoV-2 are associated with CD4+ responses, while the S, M, and some non-structural proteins contribute to CD8+ T-cell responses [16]. In this study, we found that the predicted B-cell epitopes of S (aa 407–416, aa421-427, aa1028-1049, aa1254-1273), N (aa173-189, aa235-247), M (aa163-182), and E (aa 58–68) proteins are both conserved and common in the two genetically divergent viruses (SARS-CoV and SARS-CoV-2). We predicted 16 CD8+ T-cell epitopes (five CD8+ T-cell epitopes predicted each for S and N proteins, and three each for M and E proteins) that are common to structural proteins of SARS-CoV-2 and SARS-CoV. Similarly, we found 4 CD4+ T-cell epitopes for S, 5 for N, and 2 each for M and E proteins that are common to SARS-CoV-2 and SARS-CoV, with epitope conservancy in the range of 80 to 100%. These findings imply the importance of determining the effect of pre-existing immune memory on COVID-19 disease severity. The findings of computationally predicted B- and T-cell epitopes common to both SARS-CoV-2 and SARS-CoV might have a role in heterotypic immunity against SARS-CoV-2. This hypothesis draws support from recent studies on the pre-existence of T-cell responses in a majority of healthy individuals unexposed to SARS-CoV-2 [16–18, 80]. For example, the presence of cross-reactive T cells, known in some viruses including HIV [86], is speculated to confer heterologous immunity upon exposure to a non-

identical pathogen [87, 88]. Furthermore, recent studies have experimentally demonstrated the presence of cross-reactive T cells in SARS-CoV-2 and SARS-CoV implicating the importance of heterologous immunity in SARS-CoV-2 infection [16, 18, 66].

Recently, to accelerate the process of vaccine development, researchers have employed *in silico* methods based on immunoinformatic data to develop multi-epitope vaccines without the cumbersome necessity to cultivate pathogens [27–38]. Experimental epitope identification is an expensive procedure, which presents several challenges, including antibody production to identify antigenic regions in a target protein, unavailability of animal models, and further epitope validation through crystallography. Computational approaches can help to guide experimental assays and improve precision by facilitating the selection of specific regions with a high probability of being effective epitopes [39]. An ideal multi-epitope vaccine should be designed to include epitopes that can elicit CTL, HTL, and B cells as well as induce effective responses against a targeted virus.

In this study, we designed multi-epitope chimeric constructs that included the predicted B- and T-cell epitopes of the S, E, M, and N proteins of SARS-CoV-2. Some of the epitopes predicted in this study have been verified experimentally in peer-reviewed studies. Although we included CTL, HTL, and B-cell epitopes in the chimeric constructs, these multi-epitope vaccine constructs should be subjected to further experimental validation *in vitro* and *in vivo*. The designing of efficacious multi-epitope vaccines remains a great challenge. Expression of the recombinant multi-epitope protein from the synthetic gene encoding the constructed multi-epitope vaccine can be a great challenge. Moreover, the ability of the multi-epitope construct to retain the native antigenic structure as a vaccine subunit and to elicit protective immune responses remains to be investigated. In addition, it is unclear whether the linear and conformational B-cell epitopes will fold into appropriate conformations resembling the native S, E, M, and N proteins and elicit a protective immune response. Tembusu virus (TMUV) is a newly emerging flavivirus that causes rapid egg drop and neurological symptoms in ducks. Potential *in silico* predicted B-cell epitopes of the E protein of TMUV fused with glutathione S-transferases tag have been successfully expressed in *Escherichia coli* [89]. The findings suggested that two of the four predicted B-cell epitopes could elicit the generation of neutralizing antibodies in ducks and provide protection when challenged with TMUV [89]. A chimeric recombinant antigen comprising of predicted CD4+ and CD8+ T cell-specific epitopes derived from *Leishmania infantum* was successfully expressed in *E. coli*. This purified antigen showed protective efficacy in mice against *Leishmania amazonensis* infection [90]. In our earlier study, we successfully synthesized chimeric antigens with predicted B- and T- cell epitopes of rotavirus proteins. The chimeric antigen was subsequently purified using the *E. coli* expression system, and the antigen presentation is yet to be investigated [58]. Thus, the present bioinformatic prediction provides a platform for the design of synthetic peptides and its validation using convalescent sera from patients with COVID-19. However, it remains to be investigated whether the pool of B- and T-cell epitopes identified in this study can stimulate B and T cells. The amount of cytokines released in response to these antigenic epitopes needs to be measured both *in vitro* and *in vivo*, thus providing a platform for future investigations on SARS-CoV-2-specific immune responses.

## Conclusions

In this study, the *in silico* predicted immune epitopes are limited to the structural proteins of SARS-CoV-2. Noteworthily, many of the predicted B- and T-cell epitopes, derived from multiple computational tools, have been experimentally verified in recent studies using sera from convalescent patients with COVID-19. Further rigorous experimental validations of predicted

epitopes might provide a better understanding and resolution of the immune response to SARS-CoV-2 infection.

## Supporting information

**S1 File.**
(PPTX)

## Acknowledgments

YDD is a recipient of UGC-CSIR Senior Research Fellowship (Sr. No. 2121530642, Ref. No: 20/12/2015(ii)EU-V) support for pursuing PhD work. SKR, RCD, SSS, and NDN duly acknowledged the use of computational facility at Centre for Computational Biology and Bioinformatics, Tezpur University, Assam supported by the Ministry of Science and Technology, Department of Biotechnology (DBT), Govt. of India.

## Author Contributions

**Conceptualization:** Nima D. Namsa.

**Data curation:** Himanshu Ballav Goswami, Sushmita Konwar, Chandrima Doley, Anutee Dolley, Nima D. Namsa.

**Formal analysis:** Chongtham Shyamsunder Singh, Partha Pratim Borah, Nima D. Namsa.

**Investigation:** Yengkhom Damayanti Devi, Himanshu Ballav Goswami, Sushmita Konwar, Chandrima Doley, Anutee Dolley, Arpita Devi, Dikshita Dowerah, Vashkar Biswa, Siddhartha Shankar Satapathy, Suvendra Kumar Ray, Ramesh Chandra Deka, Nima D. Namsa.

**Methodology:** Yengkhom Damayanti Devi, Himanshu Ballav Goswami, Sushmita Konwar, Chandrima Doley, Anutee Dolley, Arpita Devi, Chen Chongtham, Dikshita Dowerah, Aditya Kumar, Suvendra Kumar Ray, Robin Doley, Manabendra Mandal, Sandeep Das, Chongtham Shyamsunder Singh, Partha Pratim Borah, Nima D. Namsa.

**Project administration:** Nima D. Namsa.

**Resources:** Nima D. Namsa.

**Software:** Nima D. Namsa.

**Supervision:** Nima D. Namsa.

**Validation:** Himanshu Ballav Goswami, Sushmita Konwar, Chandrima Doley, Anutee Dolley, Chen Chongtham, Dikshita Dowerah, Vashkar Biswa, Pabitra Nath, Nima D. Namsa.

**Visualization:** Himanshu Ballav Goswami, Chandrima Doley, Arpita Devi, Dikshita Dowerah, Latonglila Jamir, Aditya Kumar, Siddhartha Shankar Satapathy, Ramesh Chandra Deka, Robin Doley, Manabendra Mandal, Sandeep Das, Pabitra Nath, Nima D. Namsa.

**Writing – original draft:** Yengkhom Damayanti Devi, Nima D. Namsa.

**Writing – review & editing:** Latonglila Jamir, Suvendra Kumar Ray, Ramesh Chandra Deka, Robin Doley, Nima D. Namsa.

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
