## [Decision Letter · Decision Letter 0]

30 Jun 2021

PONE-D-21-01974

Immunoinformatics mapping of potential epitopes in SARS-CoV-2 structural proteins

PLOS ONE

Dear Dr. Namsa,

Thank you for submitting your manuscript to PLOS ONE. After careful consideration, we feel that it has merit but does not fully meet PLOS ONE’s publication criteria as it currently stands. Therefore, we invite you to submit a revised version of the manuscript that addresses the points raised during the review process.

We look forward to receiving your revised manuscript.

Kind regards,

William C. Nierman, Ph.D.

Academic Editor

PLOS ONE

Journal Requirements:

3. Please add the tables in to the manuscript.

5. We note that Figures 2 and 3 in your submission contain copyrighted images. All PLOS content is published under the Creative Commons Attribution License (CC BY 4.0), which means that the manuscript, images, and Supporting Information files will be freely available online, and any third party is permitted to access, download, copy, distribute, and use these materials in any way, even commercially, with proper attribution. For more information, see our copyright guidelines: http://journals.plos.org/plosone/s/licenses-and-copyright.

a. You may seek permission from the original copyright holder of Figures 2 and 3 to publish the content specifically under the CC BY 4.0 license. 

Reviewers' comments:

Reviewer's Responses to Questions

**Comments to the Author**

1. Is the manuscript technically sound, and do the data support the conclusions?

Reviewer #1: Yes

2. Has the statistical analysis been performed appropriately and rigorously? 

Reviewer #1: N/A

3. Have the authors made all data underlying the findings in their manuscript fully available?

Reviewer #1: Yes

4. Is the manuscript presented in an intelligible fashion and written in standard English?

Reviewer #1: Yes

5. Review Comments to the Author

Reviewer #1: Devi and colleagues use a bioinformatic pipeline utilizing several platforms to predict B and T cell epitopes of several structural proteins (S, E, M and N) of SARS-CoV-2. The overall goal is to use this method to predict epitopes that can help guide the development of vaccines that are immunogenic, non-allergenic, and cytokine inducing. It can also present potential target for further validation for experimental work. The manuscript is generally well written, but should also address the pitfalls of the type of epitopes chosen. For example, while linear B-cell epitopes can arguably be easily identified using different approaches, it is unclear how these can elicit protective antibodies. It is difficult to design vaccines based off of linear epitopes (and arguably the same goes for conformational epitopes) – the challenge is whether expressing these epitopes on their own will truly recapitulate its native antigenic structure as a subunit vaccine. These hurdles should be discussed. In comparison, T cell epitopes are generally displayed as linear epitopes on the surface of MHC molecules and may not suffer the challenges of B cell epitopes.

1. The N protein is not expected to elicit neutralizing antibodies as it is a viral protein not expressed on the surface of viral particles. Only the S has been demonstrated to elicit neutralizing antibodies. While there may be reports of N neutralizing antibodies, it is generally accepted in the field that antibodies that target the nucleoprotein is not neutralizing. Please rephrase any statements describing neutralizing antibodies that target the N protein.

2. There are more than 3 different COVID vaccines that have received emergency use approval around the world depending on which country the authors are referring to – please clarify and correct.

3. Similarly, there are other published studies on phase 1 and 2/3 studies that reported on the safety and efficacy of other COVID vaccines other than the one the authors mentioned (ChAdOx1/AZD1222). If the authors note published studies regarding AZD1222, they should also mention ones from Moderna, Pfizer, Johnson and Johnson, etc.

4. It is not clear what the authors are trying to state in the following statement: S and N proteins are the major targets for the development of vaccines and diagnostic reagents against SARS-CoV-2 due to extended time required for performing neutralization assays and the need for BSL-3 containment facility.

5. The authors briefly mentioned conserved epitopes between SARS-CoV-1 and -CoV-2, but is not greatly discussed in the manuscript. Perhaps the authors should also put some emphasis on conserved epitopes between the two divergent viruses.

6. PLOS authors have the option to publish the peer review history of their article (what does this mean?). If published, this will include your full peer review and any attached files.

Reviewer #1: No

---

## [Author Response · Author response to Decision Letter 0]

20 Jul 2021

Revised manuscript has been submitted along to point-to-point response to both Academic and Reviewer's comments. Also please find below.

A point-to-point rebuttal to comments of editor and reviewer is appended below for your kind information and perusal. 

Journal Requirements:

Query 1. Please ensure that your manuscript meets PLOS ONE's style requirements, including those for file naming. The PLOS ONE style templates can be found at 

Response 1: We thank profusely the academic editor. We have now utilized the templates and formatted the manuscript as per requirement of PLOS ONE. 

Query 2. Please review your reference list to ensure that it is complete and correct. If you have cited papers that have been retracted, please include the rationale for doing so in the manuscript text, or remove these references and replace them with relevant current references. Any changes to the reference list should be mentioned in the rebuttal letter that accompanies your revised manuscript. If you need to cite a retracted article, indicate the article’s retracted status in the References list and also include a citation and full reference for the retraction notice.

Response 2: We now have revisited the list of references and appropriately cited all the reference in the text and the same is provided under reference list. 

Query 3. Please add the tables into the manuscript.

Response 3: As suggested all the tables have now been included at the end of the main text of revised manuscript. 

Query 4. We note that you have indicated that data from this study are available upon request. PLOS only allows data to be available upon request if there are legal or ethical restrictions on sharing data publicly. For more information on unacceptable data access restrictions, please see http://journals.plos.org/plosone/s/data-availability#loc-unacceptable-data-access-restrictions. 

Response 4a: All the data associated with the manuscript is now provided as supplementary files. 

Response 4b: All the data associated with the manuscript is now provided as supplementary files. We greatly appreciate the Journal team to update our Data Availability statement. 

Query 5. We note that Figures 2 and 3 in your submission contain copyrighted images. All PLOS content is published under the Creative Commons Attribution License (CC BY 4.0), which means that the manuscript, images, and Supporting Information files will be freely available online, and any third party is permitted to access, download, copy, distribute, and use these materials in any way, even commercially, with proper attribution. For more information, see our copyright guidelines: http://journals.plos.org/plosone/s/licenses-and-copyright.

 a. You may seek permission from the original copyright holder of Figures 2 and 3 to publish the content specifically under the CC BY 4.0 license. 

Response 5: We greatly appreciate and respect the comments of Academic Editor with regard to copyrighted images of Figures 2 and 3 and we have taken this suggestion with utmost care. The figures represented structures of SARS-CoV-2 structural proteins (S, E, M & N) which were retrieved from RCSB PDB and homology modelling was carried out by our research group. The structures were represented in surface model in Biovia Discovery studio 2017 R2 software package and the predicted B-cell epitopes were represented by different colors. The epitopes mapped on the surface were predicted using the multiple computational resources by our team and the findings are outcome of our detailed investigations. 

As suggested by the Academic Editor, we have redrawn the figure using structure of E protein with colors representing the predicted B cell epitopes reported in this manuscript. Biovia Discovery studio 2017 R2 software package have been used and the figure is provided below for your kind information and perusal. In this regard, I solicit you to reconsider this issue and provide your suggestions that might have contributed to the copyright issue. Appropriate references have been included. 

Fig. E protein of SARS-CoV-2

Reviewer’s Comments:

Comments to the Author

1. Is the manuscript technically sound, and do the data support the conclusions?

Reviewer #1: Yes

2. Has the statistical analysis been performed appropriately and rigorously?

Reviewer #1: N/A

3. Have the authors made all data underlying the findings in their manuscript fully available?

Reviewer #1: Yes

4. Is the manuscript presented in an intelligible fashion and written in standard English?

Reviewer #1: Yes

5. Review Comments to the Author

Reviewer #1: Devi and colleagues use a bioinformatic pipeline utilizing several platforms to predict B and T cell epitopes of several structural proteins (S, E, M and N) of SARS-CoV-2. The overall goal is to use this method to predict epitopes that can help guide the development of vaccines that are immunogenic, non-allergenic, and cytokine inducing. It can also present potential target for further validation for experimental work. The manuscript is generally well written, but should also address the pitfalls of the type of epitopes chosen. For example, while linear B-cell epitopes can arguably be easily identified using different approaches, it is unclear how these can elicit protective antibodies. It is difficult to design vaccines based off of linear epitopes (and arguably the same goes for conformational epitopes) – the challenge is whether expressing these epitopes on their own will truly recapitulate its native antigenic structure as a subunit vaccine. These hurdles should be discussed. In comparison, T cell epitopes are generally displayed as linear epitopes on the surface of MHC molecules and may not suffer the challenges of B cell epitopes.

Response: We profusely appreciate the suggestions of anonymous reviewer. Accordingly, we have added few lines on the above under the head_Discussion part of revised manuscript and the same is provided as follows:

In this study we designed a multi-epitope chimeric constructs that included the predicted B- and T-cell epitopes of the S, E, M and N proteins of SARS-CoV-2. Some of the epitopes predicted in this study have been verified experimentally by others. We included CTL, HTL and B-cell predicted epitopes in the chimeric constructs and designing efficacious multi-epitope vaccines remains a great challenge. Expression of the recombinant multi-epitope protein from the constructed multi-epitope vaccine gene and their ability to retain native antigenic structure as a subunit vaccine to elicit protective immune responses remains a major hurdle. In addition, it is unclear whether the linear and conformational B-cell epitopes designed to elicit a B cell will properly fold into conformations resembling the native S, E, M and N proteins to elicit a protective immune response. Tembusu virus (TMUV) is a newly emerging flavivirus that caused rapid egg drop and neurological symptoms in ducks. Potential in-silico predicted B-cell epitopes of envelope protein of TMUV fused with Glutathione S-transferases tag have been successfully expressed in E. coli [88]. The findings suggested that two of four predicted B cell epitopes could elicit neutralizing antibodies in ducks and provided protection against TMUV challenge [88]. A chimeric recombinant antigen comprising of predicted CD4+ and CD8+ T cell-specific epitopes derived from four Leishmania infantum proteins was successfully expressed and the purified antigen showed protective efficacy in mice against Leishmania amazonensis infection [89]. Thus, the present bioinformatic prediction provides signals for the design of potential synthetic peptides and its validation using convalescent sera of COVID-19 patients. However, it remains to be investigated whether a pool of B- and T-cell epitopes identified in this work can stimulate B- and T-cells by measuring the amount of released cytokines both in vitro and in vivo, thus providing a platform for future investigations. 

1. The N protein is not expected to elicit neutralizing antibodies as it is a viral protein not expressed on the surface of viral particles. Only the S has been demonstrated to elicit neutralizing antibodies. While there may be reports of N neutralizing antibodies, it is generally accepted in the field that antibodies that target the nucleoprotein is not neutralizing. Please rephrase any statements describing neutralizing antibodies that target the N protein.

Response: We profusely appreciate the suggestions of anonymous reviewer. Accordingly, we have rephrased all statements describing neutralizing antibodies that target the N protein throughout the text of revised manuscript. 

2. There are more than 3 different COVID vaccines that have received emergency use approval around the world depending on which country the authors are referring to – please clarify and correct.

Response: As suggested by the expert, we have re-written and provide all the details of COVID-19 vaccines (AstraZeneca/Oxford vaccine, Moderna, Johnson and Johnson, Pfizer/BionTech, Sinopharm, and Sinovac) which are recommended by WHO for emergency use. New references have been added from reference no. 19 to 26 under_ Introduction part of revised manuscript. 

3. Similarly, there are other published studies on phase 1 and 2/3 studies that reported on the safety and efficacy of other COVID vaccines other than the one the authors mentioned (ChAdOx1/AZD1222). If the authors note published studies regarding AZD1222, they should also mention ones from Moderna, Pfizer, Johnson and Johnson, etc.

Response: As suggested by the expert, we have re-written and provide all the details of COVID-19 vaccines (AstraZeneca/Oxford vaccine, Moderna, Johnson and Johnson, Pfizer/BionTech, Sinopharm, and Sinovac) which are recommended by WHO for emergency use. New references have been added from reference no. 19 to 26 under Introduction part of revised manuscript. 

4. It is not clear what the authors are trying to state in the following statement: S and N proteins are the major targets for the development of vaccines and diagnostic reagents against SARS-CoV-2 due to extended time required for performing neutralization assays and the need for BSL-3 containment facility.

Response: As suggested by the expert, we have re-written the above sentence as ‘S and N proteins are the major targets for the development of vaccines and diagnostic reagents against SARS-CoV-2 using these recombinant antigens. Development of rapid antibody test requires the production of recombinant antigens and its validation using the serum of convalescent COVID-19 patients in an enzyme linked immunosorbent assay (ELISA). ELISA does not require the culture of SARS-CoV-2 virus in BSL-3 containment facility [10-14,67-69]’. This has been included in the appropriate text of revised manuscript. 

5. The authors briefly mentioned conserved epitopes between SARS-CoV-1 and -CoV-2, but is not greatly discussed in the manuscript. Perhaps the authors should also put some emphasis on conserved epitopes between the two divergent viruses.

Response: We greatly appreciate the suggestions of anonymous reviewer. We now have discussed the predicted B and T cell epitopes that were common to both SARS-CoV and SARS-CoV-2 under the discussion part. 

6. PLOS authors have the option to publish the peer review history of their article (what does this mean?). If published, this will include your full peer review and any attached files.

Do you want your identity to be public for this peer review? For information about this choice, including consent withdrawal, please see our Privacy Policy.

Reviewer #1: No

While revising your submission, please upload your figure files to the Preflight Analysis and Conversion Engine (PACE) digital diagnostic tool, https://pacev2.apexcovantage.com/.

Response: We have used and uploaded figures in the PACE as suggested. 

With warm regards,

 (Nima D. Namsa)

Corresponding author

---

## [Editor Report · Decision Letter 1]

27 Jul 2021

PONE-D-21-01974R1

Immunoinformatics mapping of potential epitopes in SARS-CoV-2 structural proteins

PLOS ONE

Dear Dr. Namsa,

Thank you for submitting your manuscript to PLOS ONE. After careful consideration, we feel that it has merit but does not fully meet PLOS ONE’s publication criteria as it currently stands.

The manuscript requires an English language editorial pass to correct grammar and and other problems to bring the writing quality of the manuscript to the appropriate journal standards. 

We look forward to receiving your revised manuscript.

Kind regards,

William C. Nierman, Ph.D.

Academic Editor

PLOS ONE

Journal Requirements:

Additional Editor Comments (if provided):

I performed a careful reading of the abstract and introduction of the revised submission. My reading revealed to me that the manuscript requires an English language editorial pass to correct grammar and and other problems to bring the writing quality of the manuscript to the appropriate journal standards. I cannot be more specific on this issue nor can I review the entire submission as I cannot see line or page numbers in the manuscript and can thus not cite the location of any problems I might see in reviewing the revised submission. Please prepare a revised manuscript after the editorial pass and submit the manuscript with line and page numbers included.
---

## [Author Response · Author response to Decision Letter 1]

7 Aug 2021

A point-to-point rebuttal to comments of Academic Editor is appended below for your kind information and perusal. 

Journal Requirements:

Query 1. Please review your reference list to ensure that it is complete and correct. If you have cited papers that have been retracted, please include the rationale for doing so in the manuscript text, or remove these references and replace them with relevant current references. Any changes to the reference list should be mentioned in the rebuttal letter that accompanies your revised manuscript. If you need to cite a retracted article, indicate the article’s retracted status in the References list and also include a citation and full reference for the retraction notice.

Response 1: We gratefully acknowledged the suggestions. We now have revisited each of the cited and the listed references and corrected in the revised manuscript. 

Query 2: Additional Editor Comments (if provided): I performed a careful reading of the abstract and introduction of the revised submission. My reading revealed to me that the manuscript requires an English language editorial pass to correct grammar and and other problems to bring the writing quality of the manuscript to the appropriate journal standards. I cannot be more specific on this issue nor can I review the entire submission as I cannot see line or page numbers in the manuscript and can thus not cite the location of any problems I might see in reviewing the revised submission. Please prepare a revised manuscript after the editorial pass and submit the manuscript with line and page numbers included.

Response 2: We greatly appreciate and duly acknowledged the suggestions of Academic Editor. Accordingly, we have taken the help of an English expert service providers to correct the grammar and the writing quality of our manuscript. All the suggested changes are highlighted in the revised manuscript file with track changes. 

With warm regards,

 (Nima D. Namsa)

Corresponding author

---

## [Editor Report · Decision Letter 2]

12 Aug 2021

PONE-D-21-01974R2

Immunoinformatics mapping of potential epitopes in SARS-CoV-2 structural proteins

PLOS ONE

Dear Dr. Namsa,

Thank you for submitting your manuscript to PLOS ONE. After careful consideration, we feel that it has merit but does not fully meet PLOS ONE’s publication criteria as it currently stands. Therefore, we invite you to submit a revised version of the manuscript that addresses the points raised during the review process.

We look forward to receiving your revised manuscript.

Kind regards,

William C. Nierman, Ph.D.

Academic Editor

PLOS ONE

Journal Requirements:

Additional Editor Comments (if provided):

The English language writing quality is still not suitable for publication in PLOS ONE. Examples of the writing problems are provided by my suggested edits of the first page of the Introduction.

Line 67 deleted the comma

Line 68 add space after "NSP1."

delete comma after "ORF1b"

Line 70 replace "chymotrypsin- like" with "chymotrypsin-like"

Line 71 replace papain- like" with papain-like"

Line 76 delete comma

replace "and utilizes" with "while utilizing"

Line 85 replace ";." with "."

Line 86 replace "although a" with "A"

replace "T-cells response" with "T-cell responses"

Line 87 replace "epitopes, mainly" with "epitope"

Line 88 delete comma

delete "for"

Line 89 delete "vaccine"

I found similar kinds of problems on reading the rest of the Introduction and the first two pages of the Discussion sections: mostly inappropriate use of commas and semicolons, some misuse of singular and plural forms of nouns and verbs, incorrect presentation of a company name (BionTech vs BioNTech), failure to italicize et al., inappropriate word choice ("As on date" instead of "To date" line 481), proof reading failure (use of "previously at the beginning and end of a single sentence lines 474 and 476). These problems should be easily fixed by using commas and semicolons correctly and by careful proofreading.
---

## [Author Response · Author response to Decision Letter 2]

25 Sep 2021

Journal Requirements:

Query 1. Please review your reference list to ensure that it is complete and correct. If you have cited papers that have been retracted, please include the rationale for doing so in the manuscript text, or remove these references and replace them with relevant current references. Any changes to the reference list should be mentioned in the rebuttal letter that accompanies your revised manuscript. If you need to cite a retracted article, indicate the article’s retracted status in the References list and also include a citation and full reference for the retraction notice.

Response 1: We sincerely acknowledged the suggestions. We now have revisited each of the cited and the listed references and corrected in the revised manuscript. Importantly, citation of reference 12 have been corrected, while reference 19, 71 and 72 have been replaced with relevant current references.

Query 2: Additional Editor Comments (if provided):

The English language writing quality is still not suitable for publication in PLOS ONE. Examples of the writing problems are provided by my suggested edits of the first page of the Introduction.

Line 67 deleted the comma

Line 68 add space after "NSP1."

delete comma after "ORF1b"

Line 70 replace "chymotrypsin- like" with "chymotrypsin-like"

Line 71 replace papain- like" with papain-like"

Line 76 delete comma

replace "and utilizes" with "while utilizing"

Line 85 replace ";." with "."

Line 86 replace "although a" with "A"

replace "T-cells response" with "T-cell responses"

Line 87 replace "epitopes, mainly" with "epitope"

Line 88 delete comma

delete "for"

Line 89 delete "vaccine"

Response 2: We are grateful to Academic Editor for pointing out the grammatical errors and mistakes. All the above-mentioned suggested corrections have now been incorporated in the revised manuscript. 

Query 3: I found similar kinds of problems on reading the rest of the Introduction and the first two pages of the Discussion sections: mostly inappropriate use of commas and semicolons, some misuse of singular and plural forms of nouns and verbs, incorrect presentation of a company name (BionTech vs BioNTech), failure to italicize et al., inappropriate word choice ("As on date" instead of "To date" line 481), proof reading failure (use of "previously at the beginning and end of a single sentence lines 474 and 476). These problems should be easily fixed by using commas and semicolons correctly and by careful proofreading.

Response 3: We greatly appreciate and duly acknowledged the suggestions of Academic Editor. Accordingly, we have taken the help of an English expert service providers to correct the grammar and the writing quality of our revised manuscript. All the suggested changes are highlighted in the revised manuscript with track changes. A certificate of editing is enclosed herewith for your kind information. 

With warm regards,

 (Nima D. Namsa)

Corresponding author

---

## [Editor Report · Decision Letter 3]

4 Oct 2021

Immunoinformatics mapping of potential epitopes in SARS-CoV-2 structural proteins

PONE-D-21-01974R3

Dear Dr. Namsa,

We’re pleased to inform you that your manuscript has been judged scientifically suitable for publication and will be formally accepted for publication once it meets all outstanding technical requirements.

Kind regards,

William C. Nierman, Ph.D.

Academic Editor

PLOS ONE
---

## [Editor Report · Acceptance letter]

14 Oct 2021

PONE-D-21-01974R3 

Immunoinformatics mapping of potential epitopes in SARS-CoV-2 structural proteins 

Dear Dr. Namsa:

I'm pleased to inform you that your manuscript has been deemed suitable for publication in PLOS ONE. Congratulations! Your manuscript is now with our production department. 

Kind regards, 

on behalf of

Dr. William C. Nierman 

Academic Editor

PLOS ONE